

SciPost Phys. Lect. Notes 5 (2018)

# Efficient numerical simulations with Tensor Networks:
# Tensor Network Python (TeNPy)

**Johannes Hauschild[1⋆] and Frank Pollmann[1]**

**1** Department of Physics, TFK, Technische Universität München,
James-Franck-Straße 1, D-85748 Garching, Germany

⋆ johannes.hauschild@tum.de

## Abstract

Tensor product state (TPS) based methods are powerful tools to efficiently simulate quantum many-body systems in and out of equilibrium. In particular, the one-dimensional matrix-product (MPS) formalism is by now an established tool in condensed matter theory and quantum chemistry. In these lecture notes, we combine a compact review of basic TPS concepts with the introduction of a versatile tensor library for Python (TeNPy) [1]. As concrete examples, we consider the MPS based time-evolving block decimation and the density matrix renormalization group algorithm. Moreover, we provide a practical guide on how to implement abelian symmetries (e.g., a particle number conservation) to accelerate tensor operations.

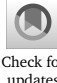
# 1   Introduction

The interplay of quantum fluctuations and correlations in quantum many-body systems can lead to exciting phenomena. Celebrated examples are the fractional quantum Hall effect [2,3], the Haldane phase in quantum spin chains [4,5], quantum spin liquids [6], and high-temperature superconductivity [7]. Understanding the emergent properties of such challenging quantum many-body systems is a problem of central importance in theoretical physics. The main difficulty in investigating quantum many-body problems lies in the fact that the Hilbert space spanned by the possible microstates grows exponentially with the system size.

To unravel the physics of microscopic model systems and to study the robustness of novel quantum phases of matter, large scale numerical simulations are essential. In certain systems where the infamous sign problem can be cured, efficient quantum Monte Carlo (QMC) methods can be applied. In a large class of quantum many-body systems (most notably, ones that involve fermionic degrees of freedom or geometric frustration), however, these QMC sampling techniques cannot be used effectively. In this case, tensor-product state based methods have been shown to be a powerful tool to efficiently simulate quantum many-body systems.

The most prominent algorithm in this context is the density matrix renormalization group (DMRG) method [8] which was originally conceived as an algorithm to study ground state properties of one-dimensional (1D) systems. The success of the DMRG method was later found to be based on the fact that quantum ground states of interest are often only slightly entangled (area law), and thus can be represented efficiently using matrix-product states (MPS) [9–11]. More recently it has been demonstrated that the DMRG method is also a useful tool to study the physics of two-dimensional (2D) systems using geometries such as a cylinder of finite circumference so that the quasi-2D problem can be mapped to a 1D one [12]. The DMRG algorithm has been successively improved and made more efficient. For example, the inclusion of Abelian and non-Abelian symmetries, [13–17], the introduction of single-site optimization with density matrix perturbation [18, 19], hybrid real-momentum space representation [20, 21], and the development of real-space parallelization [22] have increased the convergence speed and decreased the requirements of computational resources. An infinite version of the algorithm [23] has facilitated the investigation of translationally invariant systems. The success of DMRG was extended to also simulate real-time evolution allowing to study transport and non-equilibrium phenomena, [24–29]. However, the bipartite entanglement of pure states generically grows linearly with time, leading to a rapid exponential growth of the computational cost. This limits time evolution to rather short times. An exciting recent development is the generalization of DMRG to obtain highly excited states of many-body localized systems [30–32] (see also [33] for a different approach). Tensor-product states (TPS) or equivalently projected entangled pair states (PEPS), are a generalization of MPS to higher dimensions [34,35]. This class of states is believed to efficiently describe a wide range of ground states of two-dimensional local Hamiltonians. TPS serve as variational wave functions that can approximate ground states of model Hamiltonians. For this several algorithms have been proposed, including the Corner Transfer Matrix Renormalization Group Method [36], Tensor Renormalization Group (TRG) [37], Tensor Network Renormalization (TNR) [38], and loop optimizations [39].

A number of very useful review articles on different tensor network related topics appeared

over the past couple of years. Here we mention a few: Ref. [11] provides a pedagogical intro-
duction to MPS and DMRG algorithms with detailed discussions regarding their implementa-
tion. In Ref. [40], a practical introduction to tensor networks including MPS and TPS is given.
Applications of DMRG in quantum chemistry are discussed in Ref. [41].

In these lecture notes we combine a pedagogical review of basic MPS and TPS based al-
gorithms for both finite and infinite systems with the introduction of a versatile tensor library
for Python (TeNPy) [1]. In the following section, we motivate the ansatz of TPS with the area
law of entanglement entropy. In section 3 we introduce the MPS ansatz for finite systems and
explain the time evolving block decimation (TEBD) [24] and the DMRG method [8] as promi-
nent examples for algorithms working with MPS. In section 4 we explain the generalization
of these algorithms to the thermodynamic limit. For each algorithm, we give a short exam-
ple code showing how to call it from the TeNPy library. Finally, we provide a practical guide
on how to implement abelian symmetries (e.g., a particle number conservation) to accelerate
tensor operations in section 5.

## 2   Entanglement in quantum many-body systems

Entanglement is one of the fundamental phenomena in quantum mechanics and implies that
different degrees of freedom of a quantum system cannot be described independently. Over the
past decades it was realized that the entanglement in quantum many-body systems can give
access to a lot of useful information about quantum states. First, entanglement related quan-
tities provide powerful tools to extract universal properties of quantum states. For example,
scaling properties of the entanglement entropy help to characterize critical systems [42–45],
and entanglement is the basis for the classification of topological orders [46, 47]. Second, the
understanding of entanglement helped to develop new numerical methods to efficiently sim-
ulate quantum many-body systems [11, 48]. In the following, we give a short introduction to
entanglement in 1D systems and then focus on the MPS representation.

Let us consider the bipartition of the Hilbert space $\mathcal{H} = \mathcal{H}_L \otimes \mathcal{H}_R$ of a 1D system as illus-
trated in Fig. 1(a), where $\mathcal{H}_L$ ($\mathcal{H}_R$) describes all the states defined on the left (right) of a given
bond. In the so called *Schmidt decomposition*, a (pure) state $|\Psi\rangle \in \mathcal{H}$ is decomposed as

$$|\Psi\rangle = \sum_\alpha \Lambda_\alpha |\alpha\rangle_L \otimes |\alpha\rangle_R, \quad |\alpha\rangle_{L(R)} \in \mathcal{H}_{L(R)}, \tag{1}$$

where the states $\{|\alpha\rangle_{L(R)}\}$ form an orthonormal basis of (the relevant subspace of) $\mathcal{H}_L$ ($\mathcal{H}_R$)
and $\Lambda_\alpha \geq 0$. The Schmidt decomposition is unique up to degeneracies and for a normalized
state $|\Psi\rangle$ we find that $\sum_\alpha \Lambda_\alpha^2 = 1$.

An important aspect of the Schmidt decomposition is that it gives direct insight into the
*bipartite entanglement* (i.e., the entanglement between degrees of freedom in $\mathcal{H}_L$ and $\mathcal{H}_R$)
of a state, as we explain in the following. The amount of entanglement is measured by the
*entanglement entropy*, which is defined as the von-Neumann entropy $S = -\text{Tr}\left(\rho^R \log(\rho^R)\right)$ of
the reduced density matrix $\rho^R$. The *reduced density matrix* of an entangled (pure) quantum
state $|\psi\rangle$ is the density matrix of a mixed state defined on the subsystem,

$$\rho^R \equiv \text{Tr}_L\left(|\psi\rangle\langle\psi|\right). \tag{2}$$

A simple calculation shows that it has the Schmidt states $|\alpha\rangle_R$ as eigenstates and the Schmidt
coefficients are the square roots of the corresponding eigenvalues, i.e., $\rho^R = \sum_\alpha \Lambda_\alpha^2 |\alpha\rangle_R \langle\alpha|_R$
(equivalently for $\rho^L$). Hence, the entanglement entropy can be expressed in terms of the
Schmidt values $\Lambda_\alpha$,

$$S \equiv -\text{Tr}\left(\rho^R \log(\rho^R)\right) = -\sum_\alpha \Lambda_\alpha^2 \log \Lambda_\alpha^2. \tag{3}$$

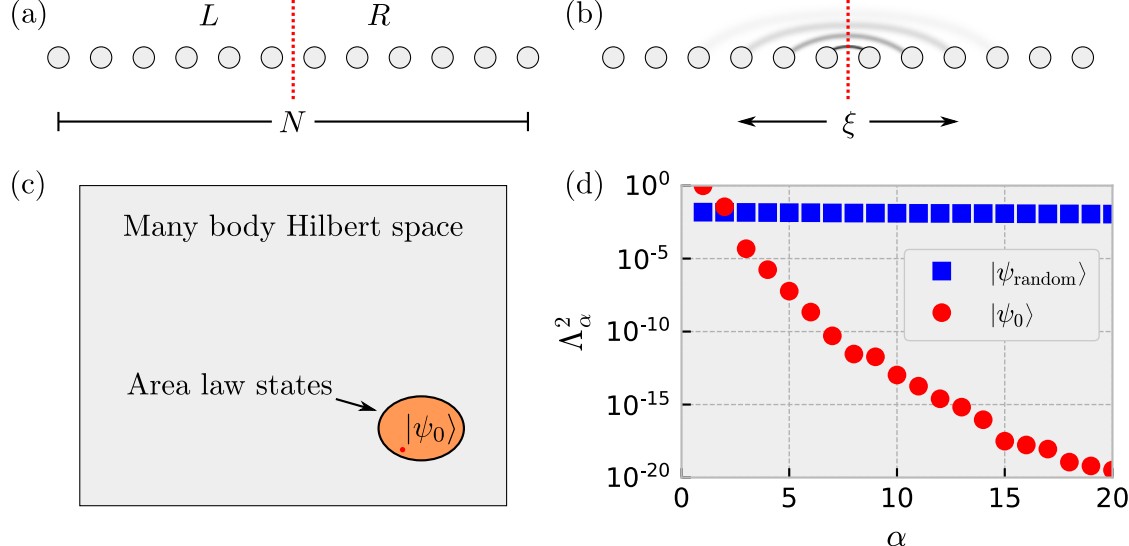

Figure 1: (a): Bipartition of a 1D system into two half chains. (b): Significant quantum fluctuations in gapped ground states occur only on short length scales. (c): 1D area law states make up a very small fraction of the many-body Hilbert space but contain all gapped ground states. (d): Comparison of the largest Schmidt values of the ground state of the transverse field Ising model ($g = 1.5$) and a random state for a system consisting of $N = 16$ spins. The index $\alpha$ labels different Schmidt values.

If there is no entanglement between the two subsystems, $S = 0$, the Schmidt decompositions consists only of a single term with $\Lambda_1 = 1$. The *entanglement spectrum* $\{\epsilon_\alpha\}$ [49] is defined in terms of the spectrum $\{\Lambda_\alpha^2\}$ of the reduced density matrix by $\Lambda_\alpha^2 = \exp(-\epsilon_\alpha)$ for each $\alpha$.

## 2.1 Area law

A "typical" state in the Hilbert space shows a *volume law*, i.e., the entanglement entropy grows proportionally with the volume of the partitions. In particular, it has been shown in Ref. [50] that in a system of $N$ sites with on-site Hilbert space dimension $d$, a randomly drawn state $|\psi_{\text{random}}\rangle$ has an entanglement entropy of $S \approx N/2 \log d - 1/2$ for a bipartition into two parts of $N/2$ sites.

In contrast, ground states $|\psi_0\rangle$ of gapped and local Hamiltonians follow instead an *area law*, i.e., the entanglement entropy grows proportionally with the area of the cut [51]. For a cut of an N-site chain as shown in Fig. 1(a) this implies that $S(N)$ is constant for $N \gtrsim \xi$ (with $\xi$ being the correlation length). This can be intuitively understood from the fact that a gapped ground state contains only fluctuations within the correlation length $\xi$ and thus only degrees of freedom near the cut are entangled, as schematically indicated in Fig. 1(b). A rigorous proof of the area law in 1D is given in Ref. [10]. In this respect, ground states are very special states and can be found within a very small corner of the Hilbert space, as illustrated in Fig. 1(c).

In slightly entangled states, only a relatively small number of Schmidt states contribute significantly. This is demonstrated in Fig. 1(d) by comparing the largest 20 Schmidt values of an area law and a volume law state for a bipartition of an $N = 16$ chain into two half chains.

As an example of an area law state, we considered here the ground state of the transverse field Ising model

$$H = -\sum_n \sigma_n^z \sigma_{n+1}^z + g\sigma_n^x, \tag{4}$$

with $\sigma_n^x$ and $\sigma_n^z$ being the Pauli operators and $g > 0$. This $\mathbb{Z}_2$ symmetric model with a quantum

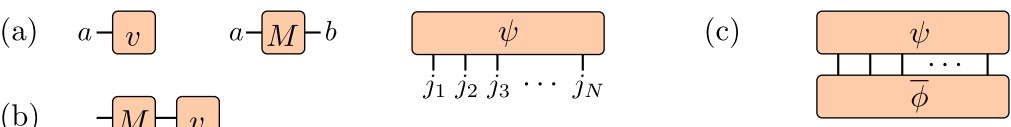

Figure 2: (a) Diagrammatic representations for a vector $v$, a matrix $M$, and the coefficients of a general wave function $|\psi\rangle = \sum_{j_1, j_2 \ldots j_N} \psi_{j_1 j_2 \ldots j_n} |j_1, j_2, \ldots, j_N\rangle$. (b) The connection of two legs symbolizes a tensor contraction, here $(Mv)_a = \sum_b M_{ab} v_b$. (c) Diagram for the overlap $\langle \phi | \psi \rangle = \sum_{j_1, j_2 \ldots j_N} \overline{\phi_{j_1 j_2 \ldots j_N}} \psi_{j_1 j_2 \ldots j_N}$ of two wave functions.

phase transition at $g_c = 1$ has two very simple limits. For $g = 0$, the ground state is twofold degenerate and given by the ferromagnetic product state (symmetry broken), and at $g \to \infty$ the ground state is a product state in which all spins are polarized by the transverse field in $x$-direction (symmetric). For intermediate values of $g$, the ground states are area law type entangled states (except at the critical point). As shown in Fig. 1(d) for a representative example of $g = 1.5$, the ground state has essentially the entire weight contained in a few Schmidt states. Generic states fulfilling the area law show a similar behavior and thus the above observation provides an extremely useful approach to compress quantum states by truncating the Schmidt decomposition. In particular, for all $\epsilon > 0$ we can truncate the Schmidt decomposition at some *finite* $\chi$ (independent of the system size) such that

$$\left\| |\psi\rangle - \underbrace{\sum_{\alpha=1}^{\chi} \Lambda_\alpha |\alpha\rangle_L \otimes |\alpha\rangle_R}_{|\psi^{\text{trunc}}\rangle} \right\| < \epsilon \tag{5}$$

This particular property of area law states is intimately related to the MPS representation of 1D quantum states, as we will discuss in the next chapter.

The situation is very different for a highly entangled (volume law) random state: All the Schmidt values are roughly constant for all $2^{N/2}$ states and thus only little weight is contained in the 20 dominant states (assuming an equal weight, we find $\Lambda_\alpha^2 \approx 1/2^{N/2}$ per Schmidt state).

## 3 Finite systems in one dimension

In this chapter, we consider a chain with $N$ sites. We label the local basis on site $n$ by $|j_n\rangle$ with $j_n = 1, \ldots, d$, e.g., for the transverse field Ising model we have spin-1/2 sites with the ($d = 2$) local states $|\uparrow\rangle, |\downarrow\rangle$. A generic (pure) quantum state can then be expanded as $|\psi\rangle = \sum_{j_1, j_2, \ldots j_N} \psi_{j_1 j_2 \ldots j_N} |j_1, j_2, \ldots, j_N\rangle$.

Before we proceed with the definition of MPS, we introduce a diagrammatic notation, which is very useful for representing tensor networks and related algorithms and has been established in the community. In this notation, a tensor with $n$ indices is represented by a symbol with $n$ legs. Connecting two legs among tensors symbolizes a tensor contraction, i.e., summing over the relevant indices. This is illustrated in Fig. 2.

## 3.1 Matrix Product States (MPS)

The class of MPS is an ansatz class where the coefficients $\psi_{j_1,\ldots,j_n}$ of a pure quantum state are decomposed into products of matrices [9, 52, 53]:

$$|\psi\rangle = \sum_{j_1,\ldots,j_N} \sum_{\alpha_2,\ldots\alpha_N} M^{[1]j_1}_{\alpha_1\alpha_2} M^{[2]j_2}_{\alpha_2\alpha_3} \ldots M^{[N]j_N}_{\alpha_N\alpha_{N+1}} |j_1, j_2, \ldots, j_N\rangle \tag{6}$$

$$\equiv \sum_{j_1,\ldots,j_N} M^{[1]j_1} M^{[2]j_2} \ldots M^{[N]j_N} |j_1, j_2, \ldots, j_N\rangle. \tag{7}$$

Here, each $M^{[n]j_n}$ is a $\chi_n \times \chi_{n+1}$ dimensional matrix, i.e., we have a set of $d$ matrices for each site, which we usually group into a tensor of order 3 as shown in Fig. 3(a). The superscript $[n]$ denotes the fact that for a generic state we have a different set of matrices on each site. The indices $\alpha_n$ of the matrices are called "bond", "virtual", or "auxiliary" indices, to distinguish them from the "physical" indices $j_n$. The matrices at the boundary are vectors, that is $\chi_1 = \chi_{N+1} = 1$, such that the matrix product in eq. (7) produces a $1 \times 1$ matrix, i.e., a single number $\psi_{j_1,\ldots,j_n}$. In that sense, the indices $\alpha_1$ and $\alpha_{N+1}$ are trivial and always 1; yet, introducing them leads to a uniform layout of the MPS such that we do not need to take special care about the boundaries in the algorithms. To become more familiar with the MPS notation, let us consider a few examples.

A **product state** $|\psi\rangle = |\phi^{[1]}\rangle \otimes |\phi^{[2]}\rangle \otimes \cdots \otimes |\phi^{[n]}\rangle$ can easily be written in the form of eq. (7): since it has no entanglement, the bond dimension is simply $\chi_n = 1$ on each bond and the $1 \times 1$ "matrices" are given by (see Fig. 3(b))

$$M^{[n]j_n} = \left(\phi^{[n]}_{j_n}\right). \tag{8}$$

Concretely, the ground state of the transverse field Ising model given in eq. (4) at large field $g \gg 1$ is close to a product state $|\leftarrow \cdots \leftarrow\rangle \equiv \left(\frac{1}{\sqrt{2}}|\uparrow\rangle - \frac{1}{\sqrt{2}}|\downarrow\rangle\right) \otimes \cdots \otimes \left(\frac{1}{\sqrt{2}}|\uparrow\rangle - \frac{1}{\sqrt{2}}|\downarrow\rangle\right)$, which we write as an MPS using the same set of matrices on each site $n$,

$$M^{[n]\uparrow} = \left(\tfrac{1}{\sqrt{2}}\right) \quad \text{and} \quad M^{[n]\downarrow} = \left(\tfrac{-1}{\sqrt{2}}\right). \tag{9}$$

For the Neel state $|\uparrow\downarrow\uparrow\downarrow\ldots\rangle$, we need different sets of matrices on odd and even sites,

$$M^{[2n-1]\uparrow} = M^{[2n]\downarrow} = \left(1\right) \quad \text{and} \quad M^{[2n-1]\downarrow} = M^{[2n]\uparrow} = \left(0\right) \tag{10}$$

for $n = 1, \ldots, N/2$.

As a first example of a state with entanglement, we consider a dimerized **product of singlets** $\left(\frac{1}{\sqrt{2}}|\uparrow\downarrow\rangle - \frac{1}{\sqrt{2}}|\downarrow\uparrow\rangle\right) \otimes \cdots \otimes \left(\frac{1}{\sqrt{2}}|\uparrow\downarrow\rangle - \frac{1}{\sqrt{2}}|\downarrow\uparrow\rangle\right)$ on neighboring sites. This state can be written with $1 \times 2$ matrices on odd sites and $2 \times 1$ matrices on even sites given by

$$M^{[2n-1]\uparrow} = \left(\tfrac{1}{\sqrt{2}} \quad 0\right), \quad M^{[2n-1]\downarrow} = \left(0 \quad \tfrac{-1}{\sqrt{2}}\right), \quad M^{[2n]\uparrow} = \begin{pmatrix} 0 \\ 1 \end{pmatrix}, \quad M^{[2n]\downarrow} = \begin{pmatrix} 1 \\ 0 \end{pmatrix}. \tag{11}$$

**Spin-1 AKLT state.** Affleck, Kennedy, Lieb, and Tasaki (AKLT) constructed an exactly solvable Hamiltonian which reads

$$H = \sum_j \vec{S}_j \vec{S}_{j+1} + \frac{1}{3}(\vec{S}_j \vec{S}_{j+1})^2 = 2 \sum_j \left(P^{S=2}_{j,j+1} - \frac{1}{3}\right) \tag{12}$$

where $\vec{S}$ are spin $S = 1$ operators and $P^{S=2}_{j,j+1}$ is a projector onto the $S = 2$ sector of the spins on sites $j$ and $j + 1$. This model is in a topologically nontrivial phase with remarkable properties of the ground state. To construct the ground state, we note that the projector $P^{S=2}_{j,j+1}$ does not

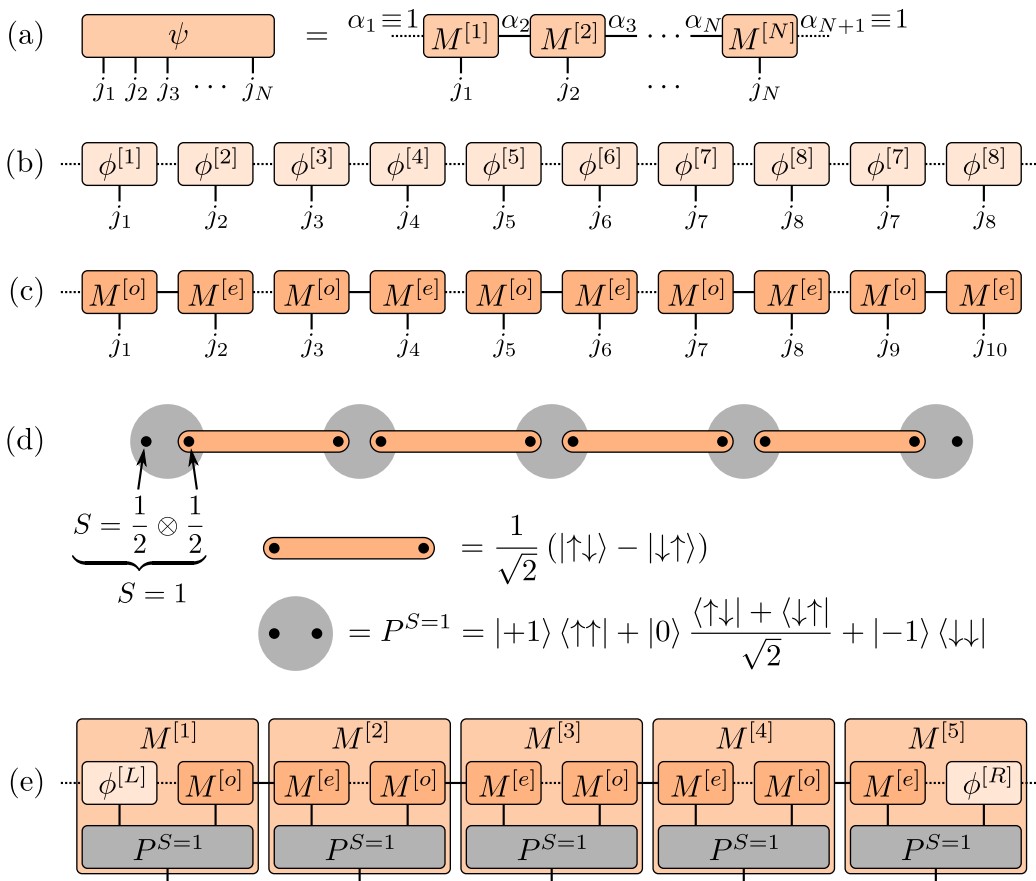

Figure 3: (a) In an MPS, the amplitude of the wave function is decomposed into a product of matrices $M^{[n]j_n}$. The indices $\alpha_1$ and $\alpha_{N+1}$ are trivial, which we indicate by dashed lines. (b) A product state can be written as a trivial MPS with bond dimensions $\chi = 1$. (c) The MPS for a product of singlets on neighboring sites, with $M^{[1]}, M^{[2]}$ given in eq. (11). (d) Diagrammatic representation of the AKLT state. The $S = 1$ sites (grey circles) are decomposed into two $S = \frac{1}{2}$ that form singlets between neighboring sites. With open boundary conditions, the $S = \frac{1}{2}$ spins on the left and right are free edge modes leading to a four-fold degeneracy of the ground state. (e) The AKLT state can be represented by an MPS with bond dimension $\chi = 2$.

give a contribution if we decompose the $S = 1$ spins on each site into two $S = \frac{1}{2}$ spins and form singlets between spins on neighboring sites, as illustrated in Fig. 3(d) [54]. While the ground state is unique on a ring with periodic boundary conditions, in a chain with open boundary conditions the $S = \frac{1}{2}$ spins on the edges do not contribute to the energy and thus lead to a 4-fold degeneracy of the ground state. Given the structure of the ground state, we can construct the corresponding MPS as shown in Fig. 3(e): We start by writing the product of singlets with the matrices of eq. 11 and add arbitrary spin-$\frac{1}{2}$ states $\phi^L$ and $\phi^R$ on the left and right. We apply the projectors $P^{S=1}$ to map the two spin-$\frac{1}{2}$ onto the physical spin-1 site, and contract the three tensors on each site to obtain the MPS structure. For sites $1 < n < N$ in the bulk, we obtain

$$M^{[n]+1} = \sqrt{\frac{4}{3}}\begin{pmatrix} 0 & 0 \\ \frac{1}{\sqrt{2}} & 0 \end{pmatrix} \qquad M^{[n]0} = \sqrt{\frac{4}{3}}\begin{pmatrix} \frac{1}{2} & 0 \\ 0 & -\frac{1}{2} \end{pmatrix} \qquad M^{[n]-1} = \sqrt{\frac{4}{3}}\begin{pmatrix} 0 & -\frac{1}{\sqrt{2}} \\ 0 & 0 \end{pmatrix}. \qquad (13)$$

Here, we included the factor $\sqrt{\frac{4}{3}}$ to normalize the MPS.

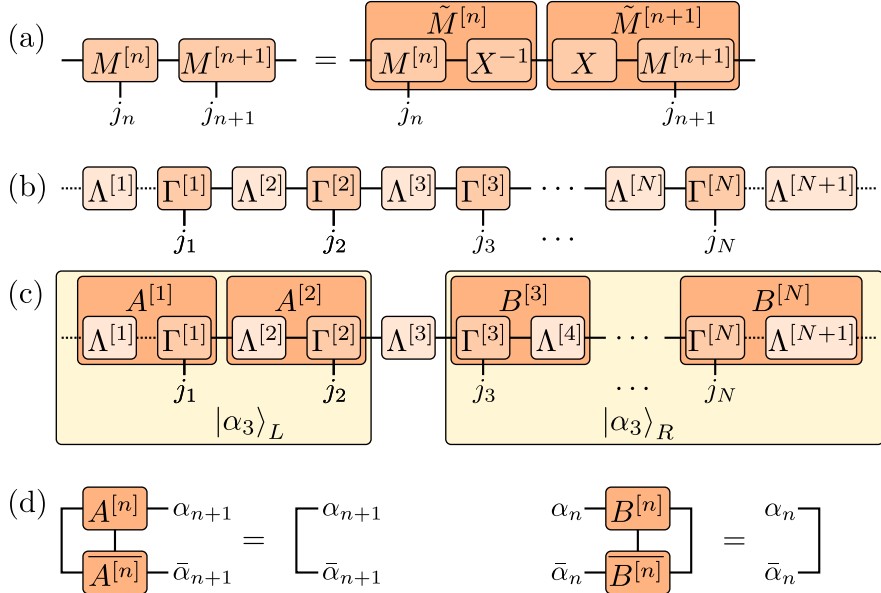

Figure 4: (a) The representation of an MPS is not unique. (b) This freedom is used to define the canonical form, where the $\Lambda^{[n]}$ are diagonal matrices containing the Schmidt values. (c) The canonical form allows to easily read off the Schmidt decomposition (1) on each bond, here exemplary on bond $n = 3$. (d) Orthonormality conditions for the Schmidt states.

In general, any state in a finite system can be decomposed *exactly* into the MPS form of eq. (7). The caveat is that for a generic state (with a volume law entanglement) the required bond dimension $\chi_{\max} := \max_n \chi_n$ increases exponentially with the number of sites $N$. However, by linking the MPS representation with the Schmidt decomposition (1), we will see that we can approximate area law states very well (in the sense of eq. (5)) by MPS with a finite bond dimension $\chi_{\max}$ [55, 56]. This link is given by the so-called canonical form of an MPS, which we introduce now.

## 3.2 Canonical form

The MPS representation (7) is not unique. Consider the bond between sites $n$ and $n + 1$, which defines a bipartition into $L = \{1, \ldots, n\}$ and $R = \{n+1, \ldots, N\}$. Given an invertible $\chi_{n+1} \times \chi_{n+1}$ matrix $X$, we can replace

$$M^{[n]j_n} \to \tilde{M}^{[n]j_n} := M^{[n]j_n} X^{-1}, \qquad M^{[n+1]j_{n+1}} \to \tilde{M}^{[n+1]j_{n+1}} := X M^{[n+1]j_{n+1}} \tag{14}$$

and still represent the same state $|\psi\rangle$, see Fig. 4(a). This freedom can be used to define a convenient "canonical form" of the MPS, following Ref. [57, 58]. Without loss of generality, we can decompose the matrices $\tilde{M}^{[n]j_n} = \tilde{\Gamma}^{[n]j_n} \tilde{\Lambda}^{[n+1]}$, where $\tilde{\Lambda}^{[n+1]}$ is a square, diagonal matrix with positive entries $\tilde{\Lambda}^{[n+1]}_{\alpha_{n+1}\alpha_{n+1}}$ on the diagonal. Performing partial contractions gives a

representation looking very similar to the Schmidt decomposition (1):

$$|\psi\rangle = \sum_{j_1,\ldots,j_N} M^{[1]j_1}\ldots M^{[n-1]j_{n-1}}\tilde{\Gamma}^{[n]j_n}\tilde{\Lambda}^{[n+1]}\tilde{M}^{[n+1]j_{n+1}}M^{[n+2]j_{n+2}}\ldots M^{[N]j_N}|j_1,\ldots,j_N\rangle$$

$$= \sum_{\tilde{\alpha}_{n+1}} \tilde{\Lambda}^{[n+1]}_{\tilde{\alpha}_{n+1}}|\tilde{\alpha}_{n+1}\rangle_L \otimes |\tilde{\alpha}_{n+1}\rangle_R, \text{ where} \tag{15}$$

$$|\tilde{\alpha}_{n+1}\rangle_L = \sum_{j_1,\ldots,j_n} \left(M^{[1]j_1}\ldots M^{[n-1]j_{n-1}}\tilde{\Gamma}^{[n]j_n}\right)_{1,\tilde{\alpha}_{n+1}}|j_1,\ldots,j_n\rangle, \tag{16}$$

$$|\tilde{\alpha}_{n+1}\rangle_R = \sum_{j_{n+1},\ldots,j_N} \left(\tilde{M}^{[n+1]j_{n+1}}M^{[n+2]j_{n+2}}\ldots M^{[N]j_N}\right)_{\tilde{\alpha}_{n+1},1}|j_{n+1},\ldots,j_N\rangle. \tag{17}$$

However, for general $M$ and $\tilde{\Gamma}^{[n]}$, the states $|\tilde{\alpha}_{n+1}\rangle_{L/R}$ will not be orthonormal. Note that we can interpret the $X$ in eq. (14) as a basis transformation of the states $|\tilde{\alpha}_{n+1}\rangle_R$ in eq. (17). The idea of the canonical form is to choose the $X$ in eq. (14) such that it maps $|\tilde{\alpha}_{n+1}\rangle_R$ to the Schmidt states $|\alpha_{n+1}\rangle_R$. Using the Schmidt values $\Lambda^{[n+1]}_{\alpha_{n+1}\alpha_{n+1}}$ on the diagonal of $\tilde{\Lambda}^{[n+1]} \to \Lambda^{[n+1]}$, we find that eq. (15) indeed gives the Schmidt decomposition. Repeating this on each bond yields the canonical form depicted in Fig. 4(b),

$$|\Psi\rangle = \sum_{j_1,\ldots,j_N} \Lambda^{[1]}\Gamma^{[1]j_1}\Lambda^{[2]}\Gamma^{[2]j_2}\Lambda^{[3]}\cdots\Lambda^{[N]}\Gamma^{[N]j_N}\Lambda^{[N+1]}|j_1,\ldots,j_N\rangle. \tag{18}$$

Here, we have introduced trivial $1 \times 1$ matrices $\Lambda^{[1]} \equiv \Lambda^{[N+1]} \equiv \left(1\right)$ multiplied to the trivial legs of the first and last tensor, again with the goal to achieve a uniform bulk. While the canonical form is useful as it allows to quickly read off the Schmidt decomposition on any bond, in practice we usually group each $\Gamma$ with one of the $\Lambda$ matrices and define

$$A^{[n]j_n} \equiv \Lambda^{[n]}\Gamma^{[n]j_n}, \qquad\qquad B^{[n]j_n} \equiv \Gamma^{[n]j_n}\Lambda^{[n+1]}. \tag{19}$$

If we write an MPS entirely with $A$ tensors ($B$ tensors), it is said to be in left (right) canonical form. In fact, all the examples given in eq. (8)-(13) are in right-canonical form. If we consider the bond between sites $n$ and $n+1$, we can write the MPS in a "mixed" canonical form with $A$ tensors up to site $n$ and $B$ tensors starting from site $n+1$, as depicted in Fig. 4(c) for $n = 2$. The $A$ and $B$ tensors transform the Schmidt basis from one bond to the next:

$$|\alpha_{n+1}\rangle_L = \sum_{\alpha_n,j_n} A^{[n]j_n}_{\alpha_n\alpha_{n+1}}|\alpha_n\rangle_L \otimes |j_n\rangle, \qquad |\alpha_n\rangle_R = \sum_{j_n,\alpha_{n+1}} B^{[n]j_n}_{\alpha_n\alpha_{n+1}}|j_n\rangle \otimes |\alpha_{n+1}\rangle_R. \tag{20}$$

Therefore, the orthonormality conditions $\langle\alpha_n|_L|\bar{\alpha}_n\rangle_L = \delta_{\alpha_n\bar{\alpha}_n} = \langle\alpha_n|_R|\bar{\alpha}_n\rangle_R$ translate into the very useful relations shown in Fig. 4(d).

One great advantage of the canonical form is that these relations allow to evaluate expectation values of local operators very easily. As shown in Fig. 5, this requires only the contraction of a few *local* tensors. If needed, we can easily convert the left and right canonical forms into each other, e.g., $A^{[n]} = \Lambda^{[n]}B^{[n]}\left(\Lambda^{[n+1]}\right)^{-1}$; since the $\Lambda^{[n]}$ are diagonal matrices, their inverses are simply given by diagonal matrices with the inverse Schmidt values[1].

As mentioned above, we can represent any state in a finite system if we allow an arbitrary bond dimension $\chi_{\max}$; but to avoid a blowup of the computational cost (exponentially in $N$), we need to *truncate* the matrices to a moderate bond dimension $\chi_{\max}$. Consider the bond between sites $n$ and $n+1$. It turns out that the simple truncation of the Schmidt decomposition is optimal in the sense of minimizing the error $\epsilon$ in eq. (5). In the (mixed) canonical form,

---

[1] If $\Lambda^{[n+1]}_{\alpha_{n+1}\alpha_{n+1}} = 0$ for some $\alpha_{n+1}$, we can remove the corresponding columns of $B^{[n]}$ and rows of $B^{[n+1]}$ before taking the inverse, as they do not contribute to the wave function.



Figure 5: Due to the orthogonality conditions depicted in Fig. 4(d), evaluating the expectation value $\langle\psi|O^{[n]}|\psi\rangle$ of a local operator $O^{[n]}$ requires only a contraction of local tensors.

we can therefore simply discard[2] some rows of $A^{[n]j_n}$, diagonal entries of $\Lambda^{[n+1]}$ and columns of $B^{[n+1]j_{n+1}}$ (namely the ones corresponding to the smallest Schmidt values). To preserve the norm of the wave function, we renormalize the Schmidt values on the diagonal of $\Lambda^{[n+1]}$ such that $\sum_{\alpha_{n+1}}\left(\Lambda^{[n+1]}_{\alpha_{n+1}\alpha_{n+1}}\right)^2 = 1$.

## 3.3 Time Evolving Block Decimation (TEBD)

In the TEBD algorithm [24], we are interested in evaluating the time evolution of a quantum state:

$$|\psi(t)\rangle = U(t)|\psi(0)\rangle. \tag{21}$$

The time evolution operator $U$ can either be $U(t) = \exp(-itH)$ yielding a real time evolution, or an imaginary time evolution $U(\tau) = \exp(-\tau H)$. The latter can be used to evaluate (finite temperature) Green's functions or as a first, conceptually simple way to find the ground state[3] of the Hamiltonian $H$ through the relation

$$|\psi_{\text{GS}}\rangle = \lim_{\tau\to\infty}\frac{e^{-\tau H}|\psi_0\rangle}{\|e^{-\tau H}|\psi_0\rangle\|}. \tag{22}$$

The TEBD algorithm makes use of the Suzuki-Trotter decomposition [59], which approximates the exponent of a sum of operators with a product of exponents of the same operators. For example, the first and second order expansions read

$$e^{(X+Y)\delta} = e^{X\delta}e^{Y\delta} + \mathcal{O}(\delta^2), \tag{23}$$

$$e^{(X+Y)\delta} = e^{X\delta/2}e^{Y\delta}e^{X\delta/2} + \mathcal{O}(\delta^3). \tag{24}$$

Here $X$ and $Y$ are operators, and $\delta$ is a small parameter. To make use of these expressions, we assume that the Hamiltonian is a sum of two-site operators of the form $H = \sum_n h^{[n,n+1]}$, where $h^{[n,n+1]}$ acts only on sites $n$ and $n+1$, and decompose it as a sum

$$H = \underbrace{\sum_{n\text{ odd}} h^{[n,n+1]}}_{H_{\text{odd}}} + \underbrace{\sum_{n\text{ even}} h^{[n,n+1]}}_{H_{\text{even}}}. \tag{25}$$

Each term $H_{\text{odd}}$ and $H_{\text{even}}$ consists of a sum of commuting operators, therefore $e^{H_{\text{odd}}\delta} = \prod_{n\text{ odd}} e^{h^{[n,n+1]}\delta}$ and similar for $H_{\text{even}}$.

We now divide the time into small time slices $\delta t \ll 1$ (the relevant time scale is in fact the inverse gap) and consider a time evolution operator $U(\delta t)$. Using, as an example, the

---

[2] Strictly speaking, this changes the Schmidt values and vectors on *other* bonds and thus destroys the canonical form! However, if the discarded weight $\sum_{\alpha>\chi}\left(\Lambda^{[n]}_{\alpha\alpha}\right)^2$ is small, this error might be ignored.

[3] As explained later on, the DMRG algorithm is a better alternative for this task.

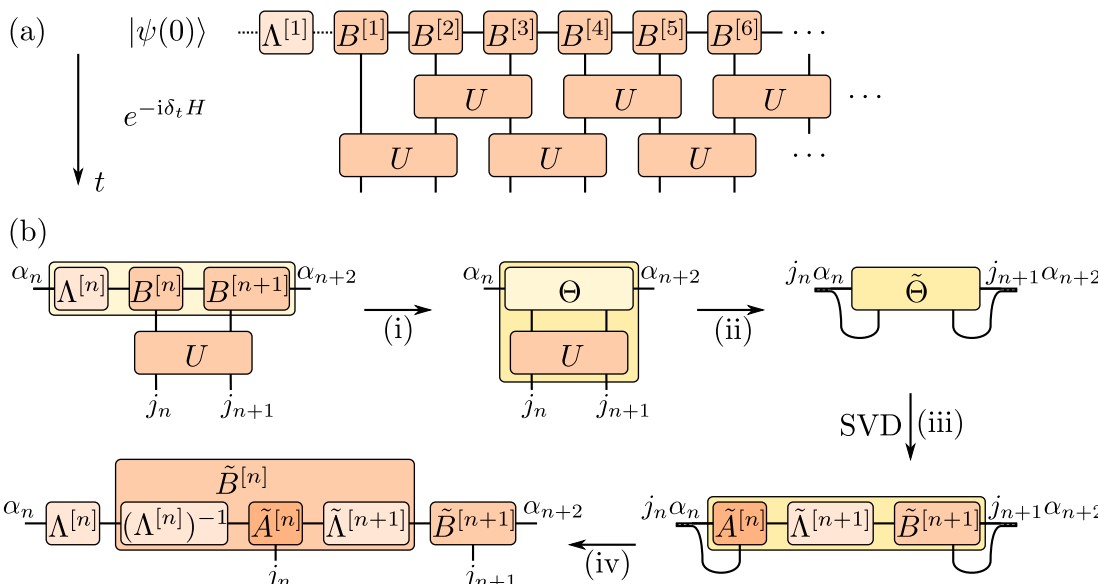

Figure 6: (a) In TEBD each time step $\delta t$ of a time evolution is approximated using a Suzuki-Trotter decomposition, i.e., the time evolution operator is expressed as a product of two-site operators. (b) Update to apply a two-site unitary $U$ and recover the MPS form, see main text for details.

first order decomposition (23), the operator $U(\delta t)$ can be expanded into products of two-site unitary operators

$$U(\delta t) \approx \left[ \prod_{n \text{ odd}} U^{[n,n+1]}(\delta t) \right]\left[ \prod_{n \text{ even}} U^{[n,n+1]}(\delta t) \right], \tag{26}$$

where $U^{[n,n+1]}(\delta t) = e^{-\mathrm{i}\delta t\, h^{[n,n+1]}}$. This decomposition of the time evolution operator is shown pictorially in Fig. 6(a). The successive application of these two-site unitary operators to an MPS is the main part of the algorithm and explained in the following.

**Local unitary updates of an MPS.** One of the advantages of the MPS representation is that local transformations can be performed efficiently. Moreover, the canonical form discussed above is preserved if the transformations are unitary [57].

A one-site unitary $U$ simply transforms the tensors $\Gamma$ of the MPS

$$\tilde{\Gamma}^{[n]j_n}_{\alpha_n \alpha_{n+1}} = \sum_{j'_n} U^{j_n}_{j'_n} \Gamma^{[n]j'_n}_{\alpha_n \alpha_{n+1}}. \tag{27}$$

In such a case the entanglement of the wave-function is not affected and thus the values of $\Lambda$ do not change.

The update procedure for a two-site unitary transformation acting on two neighboring sites $n$ and $n+1$ is shown in Fig. 6(b). We first find the wave function in the basis spanned by the left Schmidt states $|\alpha_n\rangle_L$, the local basis $|j_n\rangle$ and $|j_{n+1}\rangle$ on sites $n$ and $n+1$, and the right Schmidt states $|\alpha_{n+2}\rangle_R$, which together form an orthonormal basis $\{ |\alpha_n\rangle_L \otimes |j_n\rangle \otimes |j_{n+1}\rangle \otimes |\alpha_{n+2}\rangle_R \}$. Calling the wave function coefficients $\Theta$, the state is expressed as

$$|\psi\rangle = \sum_{\alpha_n, j_n, j_{n+1}, \alpha_{n+2}} \Theta^{j_n j_{n+1}}_{\alpha_n \alpha_{n+2}} |\alpha_n\rangle_L |j_n\rangle |j_{n+1}\rangle |\alpha_{n+2}\rangle_R. \tag{28}$$

Using the definitions of $|\alpha\rangle_{L/R}$ shown in Fig. 4(c), $\Theta$ is given by

$$\Theta^{j_n j_{n+1}}_{\alpha_n \alpha_{n+2}} = \sum_{\alpha_{n+1}} \Lambda^{[n]}_{\alpha_n \alpha_n} B^{[n],j_n}_{\alpha_n \alpha_{n+1}} B^{[n+1],j_{n+1}}_{\alpha_{n+1} \alpha_{n+2}}. \tag{29}$$

Writing the wave function in this basis is useful because it is easy to apply the two-site unitary in step (ii) of the algorithm:

$$\tilde{\Theta}^{j_n j_{n+1}}_{\alpha_n \alpha_{n+2}} = \sum_{j'_n j'_{n+1}} U^{j_n j_{n+1}}_{j'_n j'_{n+1}} \Theta^{j'_n j'_{n+1}}_{\alpha_n \alpha_{n+2}}. \tag{30}$$

Next we have to extract the new tensors $\tilde{B}^{[n]}, \tilde{B}^{[n+1]}$ and $\tilde{\Lambda}^{[n+1]}$ from the transformed tensor $\tilde{\Theta}$ in a manner that preserves the canonical form. We first "reshape" the tensor $\tilde{\Theta}$ by combining indices to obtain a $d\chi_n \times d\chi_{n+2}$ dimensional matrix $\tilde{\Theta}_{j_n \alpha_n; j_{n+1} \alpha_{n+2}}$. Because the basis $\{ |\alpha_n\rangle_L \otimes |j_n\rangle \}$ is orthonormal, as for the right, it is natural to decompose the matrix using the singular value decomposition (SVD) in step (iii) into

$$\tilde{\Theta}_{j_n \alpha_n; j_{n+1} \alpha_{n+2}} = \sum_{\alpha_{n+1}} \tilde{A}^{[n]}_{j_n \alpha_n; \alpha_{n+1}} \tilde{\Lambda}^{[n+1]}_{\alpha_{n+1} \alpha_{n+1}} \tilde{B}^{[n+1]}_{\alpha_{n+1}; j_{n+1} \alpha_{n+2}}, \tag{31}$$

where $\tilde{A}^{[n]}, \tilde{B}^{[n+1]}$ are isometries and $\tilde{\Lambda}^{[n+1]}$ is a diagonal matrix. Indeed, the suggestive notation that the new tensors are in mixed canonical form is justified, since the SVD yields a Schmidt decomposition of the wave function for a bipartition at the bond between sites $n$ and $n+1$. The isometry $\tilde{A}^{[n]}$ relates the new Schmidt states $|\alpha_{n+1}\rangle_L$ to the combined bases $|\alpha_n\rangle_L \otimes |j_n\rangle$. Analogously, the Schmidt states for the right site are obtained from the matrix $B^{[n+1]}$. Thus the diagonal matrix $\tilde{\Lambda}^{[n+1]}$ contains precisely the Schmidt values of the transformed state. In a last step (iv), we reshape the obtained matrices $\tilde{A}^{[n]}, \tilde{B}^{[n+1]}$ back to tensors with 3 indices and recover the right canonical form by

$$\tilde{B}^{[n]j_n}_{\alpha_n \alpha_{n+1}} = (\Lambda^{[n]})^{-1}_{\alpha_n \alpha_n} \tilde{A}^{[n]}_{j_n \alpha_n; \alpha_{n+1}} \tilde{\Lambda}^{[n+1]}_{\alpha_{n+1} \alpha_{n+1}} \quad \text{and} \quad \tilde{B}^{[n+1]j_{n+1}}_{\alpha_{n+1} \alpha_{n+2}} = \tilde{B}^{[n+1]}_{\alpha_{n+1}; j_{n+1} \alpha_{n+2}}. \tag{32}$$

After the update, the new MPS is still in the canonical form. The entanglement at the bond $n, n+1$ has changed and the bond dimension increased to $d\chi$. Thus the amount of information in the wave function grows exponentially if we successively apply unitaries to the state. To overcome this problem, we perform an approximation by fixing the maximal number of Schmidt terms to $\chi_{max}$. In each update, only the $\chi_{max}$ most important states are kept in step (iii), i.e., if we order the Schmidt states according to their size we simply truncate the range of the index $\alpha_{n+1}$ in eq. (31) to be $1 \ldots \chi_{max}$. This approximation limits the dimension of the MPS and the tensors $B$ have at most a dimension of $\chi_{max} \times d \times \chi_{max}$. Given that the truncated weight is small, the normalization conditions for the canonical form will be fulfilled to a good approximation. In order to keep the wave function normalized, one should divide by the norm after the truncation, i.e., divide by $\mathcal{N} = \sqrt{\sum_{j_n, j_{n+1}, \alpha_n, \alpha_{n+2}} \left| \Theta^{j_n j_{n+1}}_{\alpha_n \alpha_{n+2}} \right|^2}$.

Generically, the entanglement entropy increases with time and hence would require exponentially growing bond dimensions for an accurate description. With a finite $\chi_{max}$ limited by computational resources, the truncation errors become more severe at intermediate to large times, and the approximations made in TEBD are no longer controlled: the simulation "breaks down". For example, TEBD does not even preserve the energy when the truncation is large. An improved algorithm based on the time dependent variational principle (TDVP) was introduced in Refs. [28, 29] which performs a unitary evolution in the space of MPS with given bond dimension $\chi_{max}$.

If we perform an imaginary time evolution of the state, the operator $U$ is not unitary and thus it does not conserve the canonical form. It turns out, however, that the successive

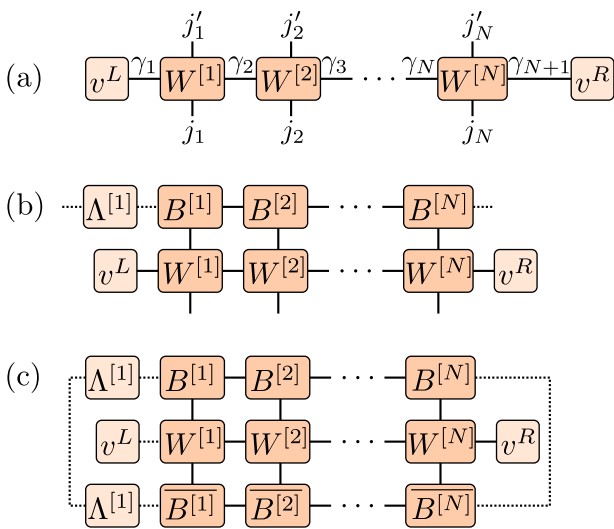

Figure 7: (a) An operator $O$ acting on the entire chain expressed as an MPO. (b) An MPO acting on an MPS in right canonical form, $O|\psi\rangle$. (c) The expectation value $\langle\psi|O|\psi\rangle$.

Schmidt decompositions assure a good approximation as long as the time steps are chosen small enough. One way to obtain very accurate results is to decrease the size of the time steps successively [58].

The simulation cost of of the TEBD algorithm scales as $\mathcal{O}(d^3\chi_{max}^3)$ and the most time consuming part of the algorithm is the SVD in step (iii). Numerically, the algorithm can become unstable when the values of $\Lambda$ become very small since the matrix has to be inverted in order to extract the new tensors in step (iv) of the algorithm. This problem can be avoided by applying a slightly modified version of this algorithm as introduced by Hastings in Ref. [60]. The following example code uses the TeNPy library [1] to perform an imaginary time evolution, finding the ground state of the transverse field Ising model (4). For a real time evolution, one can use `eng.run()` instead of `eng.run_GS()`. Note that the TEBD algorithm is rather slow in finding the ground state (especially near critical points). Moreover, in the above formulation, it can only be applied to Hamiltonians with nearest-neighbor couplings[4]. The DMRG algorithm discussed in the following represents a more efficient and versatile algorithm to study ground state properties.

```
from tenpy.networks.mps import MPS
from tenpy.models.tf_ising import TFIChain
from tenpy.algorithms import tebd

M = TFIChain({"L": 16, "J": 1., "g": 1.5, "bc_MPS": "finite"})
psi = MPS.from_product_state(M.lat.mps_sites(), [0]*16, "finite")
tebd_params = {"order": 2, "delta_tau_list": [0.1, 0.001, 1.e-5],
    "max_error_E": 1.e-6,
    "trunc_params": {"chi_max": 30, "svd_min": 1.e-10}}
eng = tebd.Engine(psi, M, tebd_params)
eng.run_GS()  # imaginary time evolution with TEBD
print("E =", sum(psi.expectation_value(M.H_bond[1:])))
print("final bond dimensions: ", psi.chi)
```

### 3.4 Matrix Product Operators (MPO)

The DMRG algorithms explained in the next section relies on expressing the Hamiltonian in the form of a matrix product operator (MPO). An MPO is a natural generalization of an MPS to the space of operators, given by

$$O = \sum_{\substack{j_1,\dots,j_N \\ j_1',\dots,j_N'}} v^L W^{[1]j_1 j_1'} W^{[2]j_2 j_2'} \cdots W^{[N]j_N j_N'} v^R |j_1,\dots,j_N\rangle \langle j_1',\dots,j_N'|, \tag{33}$$

where $W^{[n]j_n j_n'}$ are $D \times D$ matrices, and $|j_n\rangle$, $|j_n'\rangle$ represent the local basis states at site $n$, as before. At the boundaries we initiate and terminate the MPO by the left and right vectors $v^L$, $v^R$. A diagrammatic representation of an MPO is given in Fig. 7(a). The advantage of the MPO is that it can be applied efficiently to a matrix product state as shown in Fig. 7(b).

All local Hamiltonians with only short range interactions can be represented *exactly* using an MPO of a small dimension $D$. Let us consider, for example, the MPO of the anisotropic Heisenberg (XXZ) model in the presence of a field $h_n$ which can vary from site to site. The Hamiltonian is

$$H^{\text{XXZ}} = J \sum_n \left( S_n^x S_{n+1}^x + S_n^y S_{n+1}^y + \Delta S_n^z S_{n+1}^z \right) - \sum_n h_n S_n^z, \tag{34}$$

where $S_n^\gamma$, with $\gamma = x, y, z$, is the $\gamma$-component of the spin-$S$ operator at site $n$, $\Delta$ is the XXZ anisotropic interaction parameter. Expressed as a tensor product, the Hamiltonian takes the following form:

$$\begin{aligned}
H^{\text{XXZ}} = J\big( \quad & S^x \otimes S^x \otimes \mathbb{1} \otimes \cdots \otimes \mathbb{1} + \mathbb{1} \otimes S^x \otimes S^x \otimes \cdots \otimes \mathbb{1} + \dots \\
+ \quad & S^y \otimes S^y \otimes \mathbb{1} \otimes \cdots \otimes \mathbb{1} + \mathbb{1} \otimes S^y \otimes S^y \otimes \cdots \otimes \mathbb{1} + \dots \\
+ & \Delta S^z \otimes S^z \otimes \mathbb{1} \otimes \cdots \otimes \mathbb{1} + \dots \big) \\
- & h_1 S^z \otimes \mathbb{1} \otimes \mathbb{1} \otimes \cdots \otimes \mathbb{1} - \mathbb{1} \otimes h_2 S^z \otimes \mathbb{1} \otimes \cdots \otimes \mathbb{1} - \dots
\end{aligned} \tag{35}$$

The corresponding MPO has a dimension $D = 5$ and is given by

$$W^{[n]} = \begin{pmatrix} \mathbb{1} & S^x & S^y & S^z & -h_n S^z \\ 0 & 0 & 0 & 0 & JS^x \\ 0 & 0 & 0 & 0 & JS^y \\ 0 & 0 & 0 & 0 & J\Delta S^z \\ 0 & 0 & 0 & 0 & \mathbb{1} \end{pmatrix}, \tag{36}$$

where the entries of this "matrix" are operators acting on site $n$, corresponding to the indices $j_n, j_n'$, and

$$v^L = \begin{pmatrix} 1, & 0, & 0, & 0, & 0 \end{pmatrix}, \qquad\qquad v^R = \begin{pmatrix} 0, & 0, & 0, & 0, & 1 \end{pmatrix}^T. \tag{37}$$

By multiplying the matrices (and taking tensor products of the operators), one can easily see that the product of the matrices does in fact yield the Hamiltonian (35). Further details of the MPO form of operators can be found in Refs. [11, 62].

To derive the form of the matrices for a more complicated Hamiltonian, it can be useful to view the MPO as a finite state machine [63, 64]. Using this concept, the generation of an MPO for models with finite-range (two-body) interactions is automated in TeNPy [1]. The following example code creates a model representing eq. (34). Moreover, various models (including the

---

[4] One can extend TEBD for Hamiltonians with (limited) long-range couplings (e.g., next-to-nearest-neighbor couplings) by introducing so-called swap gates [61].

Heisenberg bosonic and fermionic models on cylinders and stripes) are already predefined under `tenpy.models` and can easily be generalized; in fact, the model defined below is a special case of the more general spin chain model in `tenpy.models.spin`.

```
from tenpy.models.lattice import Chain
from tenpy.networks.site import SpinSite
from tenpy.models.model import CouplingModel, \
    NearestNeighborModel, MPOModel

class XXZChain(CouplingModel, NearestNeighborModel, MPOModel):
  def __init__(self, L=2, S=0.5, J=1., Delta=1., hz=0.):
    # use predefined local Hilbert space and onsite operators
    site = SpinSite(S=S, conserve=None)
    lat = Chain(L, sites, bc="open", bc_MPS="finite") # define geometry
    CouplingModel.__init__(self, lat)
    # add terms of the Hamiltonian;
    # operators "Sx", "Sy", "Sz" are defined by the SpinSite
    self.add_coupling(J, 0, "Sx", 0, "Sx", 1)
    self.add_coupling(J, 0, "Sy", 0, "Sy", 1)
    self.add_coupling(J*Delta, 0, "Sz", 0, "Sz", 1)
    # for site dependent prefactors the strength can be a numpy array
    self.add_onsite(-hz, 0, "Sz")
    # finish initialization
    MPOModel.__init__(self, lat, self.calc_H_MPO())
    NearestNeighborModel.__init__(self, lat, self.calc_H_bond())
```

## 3.5 Density Matrix Renormalization Group (DMRG)

We now discuss the Density Matrix Renormalization Group (DMRG) algorithm [8]. Unlike TEBD, the DMRG is a variational approach to optimize the MPS, but the algorithms have many steps in common. One advantage of the DMRG is that it does not rely on a Suzuki-Trotter decomposition of the Hamiltonian and thus applies to systems with longer range interactions. We assume only that the Hamiltonian has been written as an MPO. Secondly, the convergence of the DMRG method to the ground state is in practice much faster. This is particularly the case if the gap above the ground state is small and the correlation length is long.

The schematic idea for the DMRG algorithm is as follows (see Fig. 8). Like in TEBD, the state at each step is represented by an MPS. We variationally optimize the tensors of two neighboring sites (say $n$ and $n+1$) to minimize the ground state energy $\langle \psi | H | \psi \rangle$, while keeping the rest of the chain fixed. To do so, at each step we represent the initial wave function $| \psi \rangle$ using the two site tensor $\Theta^{j_n j_{n+1}}_{\alpha_n \alpha_{n+2}}$ (as previously defined in eq. (29) the TEBD section), project the Hamiltonian into the space spanned by the basis set $\{ | \alpha_n \rangle_L \otimes | j_n \rangle \otimes | j_{n+1} \rangle \otimes | \alpha_{n+2} \rangle_R \}$, and use an iterative algorithm (e.g., Lanczos) to lower the energy. Repeating this two-site update for each pair of neighboring sites, the wave function converges to the ground state. While the Trotter decomposition requires to update first all even bonds and then odd bonds, see eq. (26), in the DMRG we perform the two-site updates in a sequential order[5], starting from the left and proceeding to the right, $n = 1, 2, 3, \ldots, L-2, L-1$, and then back from right to left, $n = L-1, L-2, \ldots, 3, 2, 1$. This sequence is called a "sweep" from left to right and back.

**Two-site update.** We start by describing the update of the tensors on two neighboring sites $n$ and $n+1$. Let us assume that we have the MPS in mixed canonical form as depicted in Fig. 8(a). We now want to find new $A^{[n]}, \Lambda^{[n]}, B^{[n+1]} \to \tilde{A}^{[n]}, \tilde{\Lambda}^{[n]}, \tilde{B}^{[n+1]}$ while keeping all other tensors fixed. Step (i) of the update is identical to the first step in the TEBD method:

---

[5] The two-site update is non-unitary and hence destroys the canonical form on *other* bonds. However, the sequential order (together with the properties of the SVD used in the update) ensures that the basis $\{ | \alpha_n \rangle_L \otimes | j_n \rangle \otimes | j_{n+1} \rangle \otimes | \alpha_{n+2} \rangle_R \}$ is still orthonormal.

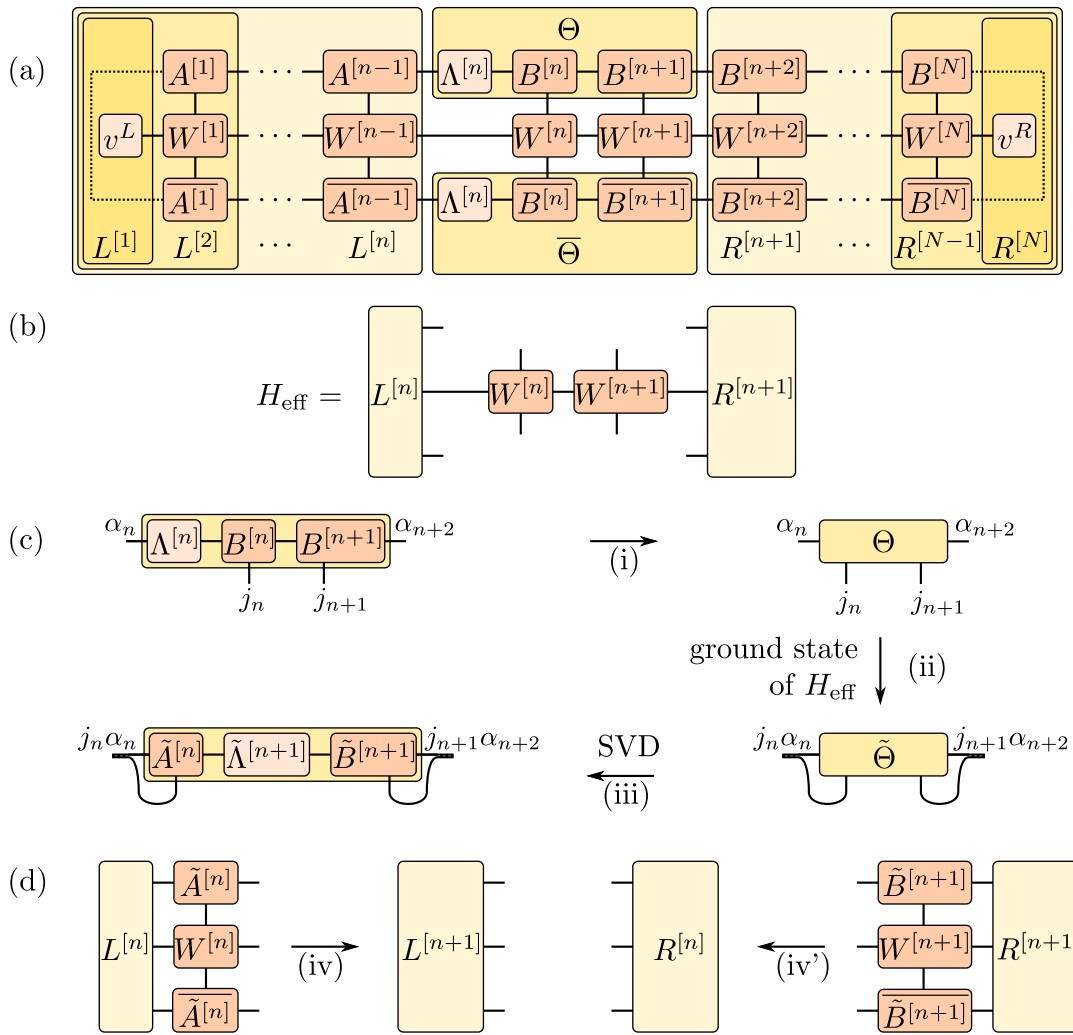

Figure 8: (a) The energy $E = \langle\psi|H|\psi\rangle$ with the MPS $|\psi\rangle$ in mixed canonical form and $H$ given by an MPO. We contract the parts to the left of site $n$ (right of site $n+1$) into the left (right) environment $L^{[n]}$ ($R^{[n+1]}$). (b) The effective Hamiltonian $H^{\mathrm{eff}}$ to update sites $n, n+1$ is the MPO projected onto the basis $\{\,|\alpha_n\rangle_L \otimes |j_n\rangle \otimes |j_{n+1}\rangle \otimes |\alpha_{n+2}\rangle_R\,\}$. (c) Update steps for the sites $n, n+1$, see main text. (d) The update rules for the environment follow from the definition in (a).

We contract the tensors for two neighboring sites to obtain the initial two-site wave function $\Theta^{j_n j_{n+1}}_{\alpha_n \alpha_{n+2}}$. The orthonormal basis $\{\,|\alpha_n\rangle_L \otimes |j_n\rangle \otimes |j_{n+1}\rangle \otimes |\alpha_{n+2}\rangle_R\,\}$ spans the variational space $|\tilde\psi\rangle = \sum_{\alpha_n, j_n, j_{n+1}, \alpha_{n+2}} \tilde\Theta^{j_n j_{n+1}}_{\alpha_n \alpha_{n+2}} |\alpha_n j_n j_{n+1} \alpha_{n+2}\rangle$ of the update, in which we must minimize the energy $E = \langle\tilde\psi|H^{\mathrm{eff}}|\tilde\psi\rangle$ in order to determine the new optimal $\tilde\Theta$. Here, $H^{\mathrm{eff}}$ is the Hamiltonian projected onto the variational space. Recall from Fig. 4(c) that the product $A^{[1]}A^{[2]}\cdots A^{[n-1]}$ gives exactly the projection from $|i_1 i_2 \ldots i_{n-1}\rangle$ to $|\alpha_n\rangle_L$, and similarly $B^{[n+2]}\cdots B^{[L]}$ maps $|i_{n+2}\ldots i_N\rangle$ to $|\alpha_{n+1}\rangle_R$. Hence, $H^{\mathrm{eff}}$ is given by the network shown in Fig. 8(b). For convenience, we have contracted the tensors strictly left of site $n$ to form $L^{[n]}$, and the ones to the right of site $n+1$ into $R^{[n+1]}$, respectively. We call these partial contractions $L^{[n]}$ and $R^{[n+1]}$ the left and right "environments". Each environment has three open legs, e.g., $L^{[n]}$ has an MPO bond index $\gamma_n$ and the two bond indices $\alpha_n, \overline\alpha_n$ of the ket and bra MPS. For now let us assume that we already performed these contractions; we will later come back to the initialization of them.

Grouping the indices on the top and bottom, we can view $H^{\text{eff}}$ as a matrix with dimensions up to $\chi^2_{\max} d^2 \times \chi^2_{\max} d^2$. Minimizing the energy $E = \langle \tilde{\psi} | H^{\text{eff}} | \tilde{\psi} \rangle$ thus means to find the the $\chi^2_{\max} d^2$ dimensional ground-state vector $\tilde{\Theta}$ of the effective Hamiltonian. Since this is the computationally most expensive part of the DMRG algorithm, it is advisable to use an iterative procedure like the Lanczos algorithm instead of a full diagonalization of $H^{\text{eff}}$. If the previous two-site wave function $\Theta$ obtained in step (i) is already a good approximation of the ground state, the Lanczos algorithm typically converges after a few steps and thus requires only a few "matrix-vector" multiplications, i.e., contractions of $H^{\text{eff}}$ with $\Theta$. Note that the scaling of such a matrix-vector multiplication is better (namely $\mathcal{O}(\chi^3_{\max} D d^2 + \chi^2_{\max} D^2 d^3)$) if we contract the tensors $L^{[j]}, W^{[n]}, W^{[n+1]}, R^{[n+2]}$ one after another to $\Theta$, instead of contracting them into a single tensor and applying it to $\Theta$ at once (which would scale as $\mathcal{O}(\chi^4_{\max} d^4)$).

This update step can be compared to the TEBD update where we obtain a new wave-function $\tilde{\Theta}$ after applying an time-evolution operator. As with TEBD, we split the new $\tilde{\Theta}$ using an SVD in step (iii), and must truncate the new index $\alpha_{n+1}$ to avoid a growth $\chi \to d\chi$ of the bond dimension. It is important that the left and right Schmidt basis $|\alpha_n\rangle_L, |\alpha_{n+2}\rangle_R$ are orthonormal, on one hand to ensure that the eigenstate of $H^{\text{eff}}$ (seen as a matrix) with the lowest eigenvalue indeed minimizes $E = \langle \tilde{\psi} | H^{\text{eff}} | \tilde{\psi} \rangle$ and on the other hand to ensure an optimal truncation at the given bond. Assuming that this is the case, the isometry properties of the SVD matrices imply that the orthonormality conditions also hold for the updated Schmidt states $|\alpha_n\rangle_{L/R}$ defined about the central bond.

At this point, we have improved guesses for the tensors $\tilde{A}^{[n]}, \tilde{\Lambda}^{[n+1]}, \tilde{B}^{[n]}$ (after a reshaping into the desired form) and can move on to the next bond. Note that we moved the center of the mixed canonical form to the central bond $n : n+1$. If we move to the right, the next two-site wave function $\Theta$ for step (i) is thus again given by $\tilde{\Lambda}^{[n+1]}\tilde{B}^{[n+1]}B^{[n+2]}$, while if we move to the left, we need to use $A^{[n-1]}\tilde{A}^{[n]}\tilde{\Lambda}^{[n+1]}$. Moreover, we need to find the next environments.

The starting environments on the very left and right are simply given by (see Fig. 8(a))

$$L^{[1]}_{\alpha_1 \overline{\alpha}_1 \gamma_1} = \delta_{\alpha_1 \overline{\alpha}_1} v^L_{\gamma_1}, \qquad\qquad R^{[N]}_{\alpha_{N+1} \overline{\alpha}_{N+1} \gamma_{N+1}} = \delta_{\alpha_{N+1} \overline{\alpha}_{N+1}} v^R_{\gamma_{N+1}}. \qquad (38)$$

Here, the $\delta_{\alpha_1 \overline{\alpha}_1}$ and $\delta_{\alpha_{N+1} \overline{\alpha}_{N+1}}$ are trivial since $\alpha_1$ and $\alpha_{N+1}$ are dummy indices which take only a single value. The other environments can be obtained from a simple recursion rule shown as step (iv) of Fig. 8(d). Using this recursion rule, $R^{[2]}$ required for the first update of the sweep can be obtained by an iteration starting from the right-most $R^{[N]}$. Note that the update on sites $n, n+1$ does not change the right environments $R^k$ for $k > n+1$. Thus it is advisable to keep the environments in memory, such that we only need to recalculate the left environments when sweeping from left to right, and vice versa in the other direction.

The procedure described above optimizes always two sites at once. Ref. [18] introduced a way to perturb the density matrices during the algorithm. This allows to perform DMRG while optimizing only a single site at once, called "single-site DMRG" or "1DMRG" in the literature, and helps to avoid getting stuck in local minima. A detailed discussion of two-site vs. single-site DMRG and a improved version of the density matrix perturbation can be found in Ref. [19]. Especially for models with long-range interactions (which appear for example when mapping a quasi-2D cylinder to a 1D chain) or models with topological phases, this density matrix perturbation can be necessary to converge towards the correct ground state. In TeNPy, this perturbation of Ref. [18] can be activated with the parameter `"mixer"`; the single-site DMRG is (at the moment) not implemented.

As noted above, DMRG is usually faster and more stable than an imaginary time evolution with TEBD. Adapting the TeNPy code from the TEBD section to run DMRG instead of TEBD is very simple [1]:

```
from tenpy.networks.mps import MPS
```

```
from tenpy.models.tf_ising import TFIChain
from tenpy.algorithms import dmrg

M = TFIChain({"L": 16, "J": 1., "g": 1.5, "bc_MPS": "finite"})
psi = MPS.from_product_state(M.lat.mps_sites(), [0]*16, "finite")
dmrg_params = {"trunc_params": {"chi_max": 30, "svd_min": 1.e-10}}
dmrg.run(psi, M, dmrg_params)  # find the ground state
print("E =", sum(psi.expectation_value(M.H_bond[1:])))
print("final bond dimensions: ", psi.chi)
```

## 4  Infinite systems in one dimension

For translation invariant systems, we can take the thermodynamic limit in which the number of sites $N \to \infty$, generalizing (7) to

$$|\psi\rangle = \sum_{\dots j_{n-1}, j_n, j_{n+1}, \dots} \cdots M^{[n-1]j_{n-1}} M^{[n]j_n} M^{[n+1]j_{n+1}} \cdots |\dots, j_{n-1}, j_n, j_{n+1}, \dots\rangle. \tag{39}$$

We can ensure the translation invariance of this infinite MPS (iMPS) by construction if we simply take all the tensors $M^{[n]} \to M$ in eq. (39) to be the same [also called uniform MPS (uMPS) in the literature]. The paramagnetic product state $|\cdots \leftarrow\leftarrow \cdots\rangle$ with the tensors of eq. (9) is a trivial example for such a translation invariant state; another example is the AKLT state given in eq. (13). In general, we might only have a translation invariance by shifts of (multiples of) $L$ sites. In this case we introduce a repeating unit cell of $L$ sites with $L$ different tensors, $M^{[n]} = M^{[n+L]}$ in eq. (39). For example, the Neel state $|\cdots \uparrow\downarrow\uparrow\downarrow \cdots\rangle$ is only invariant under a translation by (multiples of) $L = 2$ sites, with the tensors on even and odd sites as given in eq. (10) for the finite case, illustrated in Fig. 9(a). The length $L$ of the unit cell should be chosen compatible with the translation symmetry of the state to be represented, e.g., for the Neel state $L$ should be a multiple of 2. Choosing $L$ larger than strictly necessary allows to check the translation invariance explicitly.

At first sight, it might seem that we need to contract an infinite number of tensors to evaluate expectation values of local operators, as the corresponding network consists of an infinite number of tensors. However, as shown in Fig. 9(b) for a unit cell of $L = 2$ sites, the network has a repeating structure consisting of the so-called *transfer matrix* $T$ defined as

$$T_{\alpha\bar{\alpha},\gamma\bar{\gamma}} = \sum_{j_1, j_2, \beta, \bar{\beta}} M^{[1]j_1}_{\alpha\beta} \overline{M^{[1]j_1}_{\bar{\alpha}\bar{\beta}}} M^{[2]j_2}_{\beta\gamma} \overline{M^{[2]j_2}_{\bar{\beta}\bar{\gamma}}}. \tag{40}$$

A state is called *pure* if the largest (in terms of absolute value) eigenvalue of $T$ is unique and *mixed* if it is degenerate. In the following, we will always assume that the state is pure (in fact every mixed state can be uniquely decomposed into a sum of pure ones). We renormalize the iMPS such that the largest eigenvalue of $T$ is 1. The eigenvector depends on the gauge freedom of eq. (14), which we can use to bring the iMPS into the convenient canonical form defined by the Schmidt decomposition on each bond, see Fig. 9(c). An algorithm to achieve this is described in Ref. [65]. For an iMPS in right-canonical form, i.e., $M^{[n]j_n} \to B^{[n]j_n} \equiv \Gamma^{[n]j_n} \Lambda^{[n+1]}$, the orthonormality condition of the Schmidt vectors depicted in Fig. 4(d) applied to the whole unit cell implies that $\delta_{\gamma\bar{\gamma}}$ is a right eigenvector of $T$ with eigenvalue 1, as depicted in Fig. 9(d). Note that $T$ is not symmetric and hence left and right eigenvectors differ; the left eigenvector to the eigenvalue 1 is $(\Lambda^{[1]}_\alpha)^2 \delta_{\alpha\bar{\alpha}}$. All other eigenvalues of the transfer matrix have magnitude smaller than 1. Therefore, the repeated application of the transfer matrix in the network of the expectation value projects onto these dominant left and right eigenvectors, and the infinite network of the expectation value $\langle\psi|O_n|\psi\rangle$ simplifies to a local network as in the finite case, see Fig. 5.

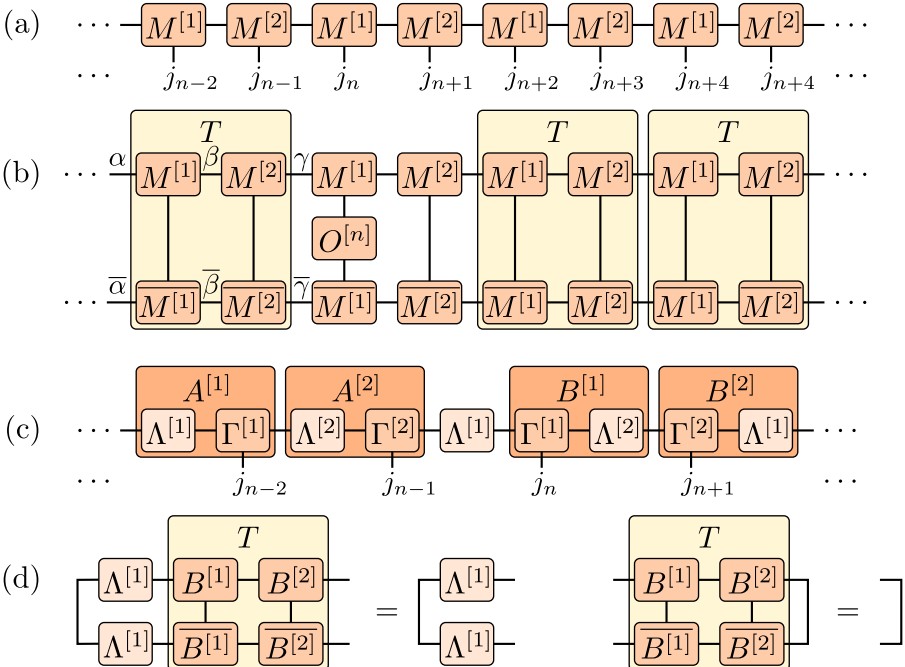

Figure 9: (a) An infinite MPS with a unit cell of $L = 2$ sites. (b) The expectation value $\langle\psi|O_n|\psi\rangle$ contains the transfer matrix $T$ as a repetitive struture. (c) The canonical form is defined as in the finite case. (d) The orthonormality conditions of the Schmidt states yield eigenvector equations for the transfer matrix.

A similar reasoning can be used for the correlation function $\langle\psi|O_n O_m|\psi\rangle$. Projecting onto the dominant eigenvectors left of $O_n$ and right $O_m$, we arrive at the network of Fig. 10(a). In between the operators $O_n$ and $O_m$, the transfer matrix $T$ appears $N = \lfloor\frac{|m-n|}{L}\rfloor - 1$ times, where $\lfloor\cdot\rfloor$ denotes rounding down to the next integer. Formally diagonalizing the transfer matrix to take the $N$-th power shows that the correlation function is a sum of exponentials,

$$\langle\psi|O_n O_m|\psi\rangle = \langle\psi|O_n|\psi\rangle\,\langle\psi|O_m|\psi\rangle + (\eta_2)^N C_2 + (\eta_3)^N C_3 + \cdots. \tag{41}$$

Here, $\eta_i$ labels the $i$-largest eigenvalue corresponding to the left and right eigenvectors $\eta_i^{[L/R]}$, $C_i = (O_n^{[L]}\eta_i^{[R]})(\eta_i^{[L]}O_n^{[R]})$ denotes the remaining parts of the network shown in Fig. 10, and we identified the $C_1 = \langle\psi|O_n|\psi\rangle\,\langle\psi|O_m|\psi\rangle$ in the term of the dominant eigenvalue $\eta_1 = 1$. The decay of the correlations is thus determined by the second largest eigenvalue $\eta_2$, which yields the correlation length

$$\xi = -\frac{L}{\log|\eta_2|}. \tag{42}$$

Numerically, it is readily obtained from a sparse algorithm finding extremal eigenvalues of $T$.

## 4.1 Infinite Time Evolving Block Decimation (iTEBD)

Generalizing TEBD to infinite systems is very simple and requires only minor modifications in the code [58]. Without loss of generality we assume that the Hamiltonian is translation invariant by $L$ sites as the iMPS; otherwise we enlarge the unit cells. As in the finite case, we use a Suzuki-Trotter decomposition to obtain the expression of the time evolution operator $U(t)$ given in eq. (26), but now the index $n$ runs over all integer numbers, $n \in \mathbb{Z}$. If we apply the two-site unitary $U^{[n,n+1]} = e^{ih^{[n,n+1]}\delta t}$ on the iMPS to update the matrices $B^{[n]}$ and $B^{[n+1]}$ as illustrated in Fig. 6(b), this corresponds due to translation invariance to the action of $U^{[n,n+1]}$

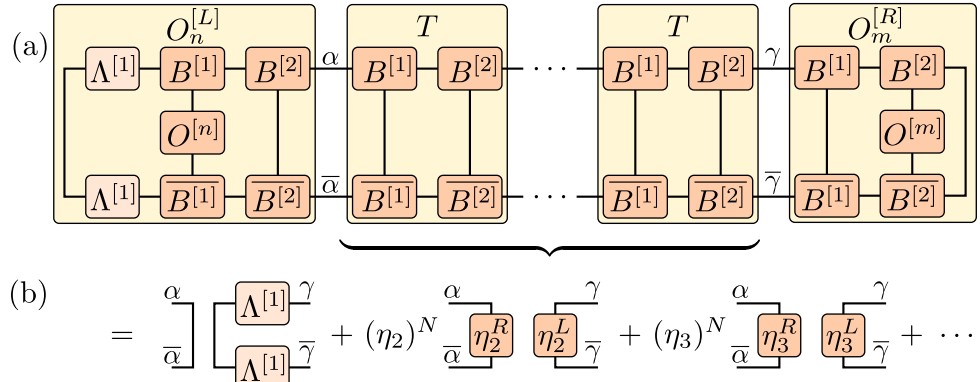

Figure 10: (a) Correlation function $\langle\psi|O_nO_m|\psi\rangle$. (b) Expansion of $T^N$ in terms of dominant eigenvectors and eigenvalues of $T$ for large $N$. The second largest eigenvalue $\eta_2$ of $T$ determines the correlation length via eq. 42.

on the sites $(n+mL, n+1+mL)$ for any $m \in \mathbb{Z}$. Therefore, we can use the same two-site update as in the finite case; the only difference is that the matrices of the iMPS represent only the unit cell with nontrivial left and right bonds, and compared to a finite system with $L$ sites we have an additional term $h^{[L,L+1]} \equiv h^{[L,1]}$ accross the boundary of the unit cell.

Note that the iTEBD algorithm is different from a time evolution in a finite system of $N = L$ sites with periodic boundary conditions. For analytical calculations with MPS in systems with periodic boundary conditions, it can be useful to change the definition of an MPS from eq. (7) to

$$|\psi\rangle = \sum_{j_1,\ldots,j_N} \mathrm{Tr}\left(M^{[1]j_1}M^{[2]j_2}\ldots M^{[N]j_N}\right)|j_1, j_2, \ldots, j_N\rangle, \qquad (43)$$

which has at first sight the same tensor network structure as an iMPS. However, cutting a single bond of such a finite MPS with periodic boundary conditions does not split it into two parts. Therefore, the canonical form (which relies on the Schmidt decomposition) is not well defined in a system with periodic boundary conditions (or in general for any tensor network state in which the bonds form loops)[6]. Since the two-site update scheme of iTEBD implicitly uses the canonical form, it implements the time evolution in the infinite system with open boundary conditions. This also becomes evident by the fact that the bond dimension $\chi_{\max}$ – in other words the number of Schmidt states taken into account – can get larger than the Hilber space dimension $d^L$ inside one unit cell.

In TeNPy one only needs to change the parameter `"bc_MPS"` from `"finite"` to `"infinite"` to switch from TEBD to iTEBD [1]. In addition, we can choose a smaller unit cell of just $L = 2$ sites and calculate the energy per site.

```
from tenpy.networks.mps import MPS
from tenpy.models.tf_ising import TFIChain
from tenpy.algorithms import tebd

M = TFIChain({"L": 2, "J": 1., "g": 1.5, "bc_MPS": "infinite"})
psi = MPS.from_product_state(M.lat.mps_sites(), [0]*2, "infinite")
tebd_params = {"order": 2, "delta_tau_list": [0.1, 0.001, 1.e-5],
    "max_error_E": 1.e-6,
    "trunc_params": {"chi_max": 30, "svd_min": 1.e-10}}
eng = tebd.Engine(psi, M, tebd_params)
eng.run_GS()  # imaginary time evolution with TEBD
print("E =", sum(psi.expectation_value(M.H_bond))/psi.L)
print("final bond dimensions: ", psi.chi)
```

---

[6]A generalization of the canonical form for networks with closed loops was recently given in Ref. [66].

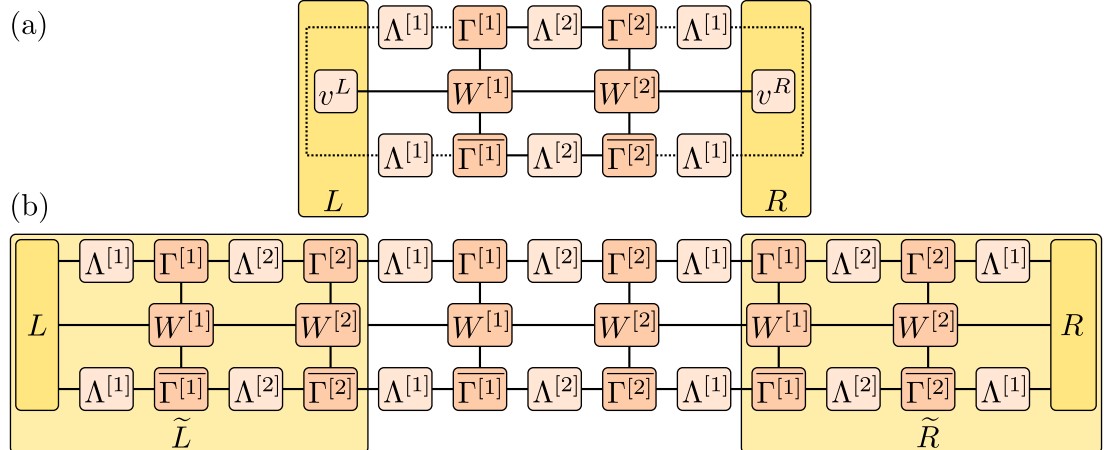

Figure 11: (a) For iDMRG (here with a unit cell of $L = 2$ sites), we initialize the environments and perform updates like in DMRG of a finite system with $L$ sites. (b) Between the sweeps, we increase the system size by inserting a unit-cell of $L$ sites into each of the environments (assuming translation invariance of the iMPS).

## 4.2 Infinite Density Matrix Renormalization Group (iDMRG)

While iTEBD works directly in the thermodynamic limit $N \rightarrow \infty$ by employing translation invariance, for the infinite Density Matrix Renormalization Group (iDRMG) one should think of a finite system with a growing number of sites - the "renormalization group" in the name refers to this. Let us assume that the Hamiltonian is given as an MPO with a translation invariant unit cell consisting of $W^{[n]}$, $n = 1, \cdots, L$, which we can terminate with the boundary vectors $v^L, v^R$ to obtain the Hamiltonian of a finite system with a multiple of $L$ sites. We initialize the environments and perform two-site updates during a sweep exactly like in finite DMRG. The crucial difference is that we increase the system size between the sweeps as illustrated in Fig. 11(b): assuming translation invariance, we redefine the left and right environments $\widetilde{L} \rightarrow L$ and $\widetilde{R} \rightarrow R$ to include additional unit cells. Moreover, we need to extend the sweep to include an update on the the sites $(L, L+1) \equiv (L, 1)$. With each unit cell inserted, the described finite system grows by $L$ sites, where we focus only only on the central $L$ sites. Full translation invariance is only recovered when the iDMRG iteration of sweeps and growing environments converges to a fix point, at which the environments describe infinite half-chains.

One subtlety of the above prescription lies in the interpretation of the energy $E$ obtained during the diagonalization step. Is it the (infinite) energy of the infinite system? Keeping track of the number of sites $\ell_{R/L}$ included into each of the environments, we see that the energy $E$ corresponds to a system of size $N = \ell_L + L + \ell_R$. By monitoring the change in $E$ with increased $N$, we can extract the energy per site. This is convenient for problems in which there is no few-site Hamiltonian with which to evaluate the energy.

When symmetry breaking is expected, it is helpful to initialize the environments by repeatedly performing the iDMRG update *without* performing the Lanczos optimization, which builds up environments using an initial symmetry broken MPS.

Like for iTEBD, the switch from DMRG to iDMRG in TeNPy is simply accomplished by a change of the parameter "bc_MPS" from "finite" to "infinite", a minimal example is given below.

```
from tenpy.networks.mps import MPS
from tenpy.models.tf_ising import TFIChain
from tenpy.algorithms import dmrg
```

```
M = TFIChain({"L": 2, "J": 1., "g": 1.5, "bc_MPS": "infinite"})
psi = MPS.from_product_state(M.lat.mps_sites(), [0]*2, "infinite")
dmrg_params = {"trunc_params": {"chi_max": 30, "svd_min": 1.e-10}}
dmrg.run(psi, M, dmrg_params)  # find the ground state
print("E =", sum(psi.expectation_value(M.H_bond))/psi.L)
print("final bond dimensions: ", psi.chi)
```

To close this chapter, we mention the varitional uniform Matrix Product State algorithm (VUMPS) as a new alternative to iDMRG, see Ref. [67] and references therein. In short, the method can preserve a strict uniform structure of the infinite MPS in a very clever way by summing up geometric series appearing in the effective Hamiltonian.

# 5   Charge conservation

If there is a unitary $U$ which commutes with the Hamiltonian, $U$ and $H$ can be diagonalized simultaneously, in other words the Hamiltonian has a block-diagonal structure when written in the eigenbasis of $U$. This can be exploited to speed up simulations: for example diagonalizing a full $N \times N$ matrix requires $\mathcal{O}(N^3)$ FLOPs, whereas the diagonalization of $m$ blocks of size $\frac{N}{m}$ requires $\mathcal{O}\left(m\left(\frac{N}{m}\right)^3\right)$ FLOPs. A similar reasoning holds for the singular value decomposition and matrix or tensor products. While exploiting the block structure does not change the scaling of the considered algorithm with the total dimension of the tensors, the gained speedup is often significant and allows more precise simulations with larger bond dimensions at the same computational cost.

For tensor networks, the basic idea is that we can ensure a block structure of *each* tensor individually. One can argue based on representation theory of groups that the tensors can be decomposed in such a block structure [14, 15]. However, here we present a bottom-up approach which is closer to the implementation. Motivated by an example, we will state a simple "charge rule" which fixes the block structure of a tensor by selecting entries which have to vanish. We explain how to define and read off the required charge values. Then we argue that tensor network algorithms (like TEBD or DMRG) require only a few basic operations on tensors, and that these operations can be implemented to preserve the charge rule (and to exploit the block strucure for the speedup).

In these notes, we focus exclusively on global, abelian symmetries which act locally in the computational basis. and refer to Refs. [13,14,16,17] for the non-abelian case, which requires a change of the computational basis and is much more difficult to implement.

## 5.1   Definition of charges

For concreteness, let us now consider two spin-$\frac{1}{2}$ sites coupled by

$$H = \vec{S}_1 \cdot \vec{S}_2 = \sum_{ab} H_{ab} |a\rangle \langle b| \text{ with } H_{ab} = \frac{1}{4} \begin{pmatrix} 1 & & & \\ & -1 & 1 & \\ & 1 & -1 & \\ & & & 1 \end{pmatrix}, \tag{44}$$

where we have represented $H$ in the basis $\{|a\rangle\} \equiv \{|\uparrow\uparrow\rangle, |\uparrow\downarrow\rangle, |\downarrow\uparrow\rangle, |\downarrow\downarrow\rangle\}$ and omitted zeros. Indeed, we clearly see a block-diagonal structure in this example, which stems from the conservation of the magnetization[7] $S^z = S_1^z + S_2^z$. We can identify the blocks if we note that the

---

[7] We call this a $U(1)$ symmetry since $H$ commutes with $U = \exp(i\phi \sum_j S_j^z) = \prod_j e^{i\phi S_j^z}$ which has a $U(1)$ group structure. If one thinks of particles (e.g., Fermions after using a Jordan-Wigner transformation), this symmetry corresponds to the particle number conservation. In general, one could also exploit the non-abelian $SU(2) \cong SO(3)$ symmetry of spin rotations, but we focus on the simpler case of abelian symmetries.

$$(a) \quad \begin{array}{c} s_1 \\ s_2 \end{array}\!\!\to\!\boxed{H}\!\to\!\!\begin{array}{c} t_1 \\ t_2 \end{array} \qquad a\to\boxed{H}\to b \qquad s\to\boxed{S^z}\to t \qquad s\to\boxed{S^+}\to t \qquad s\to\boxed{S^-}\to t$$

$$a\to\boxed{v} \qquad \boxed{\overline{v}}\to b \qquad \begin{array}{c} s_1 \\ s_2 \end{array}\!\!\to\!\boxed{v}$$

$$(b) \quad l\to\boxed{A^{[1]}}\to c\to\boxed{\Lambda^{[2]}}\to c\to\boxed{B^{[2]}}\to r$$
$$\phantom{xxxxx}\uparrow s_1 \phantom{xxxxxxxxx} \uparrow s_1$$

Figure 12: (a) Diagramatic representation of the tensors in Tab. 1. We indicate the signs $\zeta$ by small arrows on the legs. (b) Sign convention for the MPS.

considered basis states are eigenstates of $S^z$ and inspect their eigenvalues: $|\uparrow\uparrow\rangle$ corresponds to the eigenvalue $\hbar$, the two states $|\uparrow\downarrow\rangle, |\downarrow\uparrow\rangle$ form a block to the eigenvalue 0, and $|\downarrow\downarrow\rangle$ corresponds to $-\hbar$. To avoid floating point errors we rescale the "charges" to take only integer values; clearly, whenever $S^z$ is conserved, so is $q := 2S^z/\hbar$, but the latter takes the simple values 2, 0 and $-2$ for the four basis states $|a\rangle$ considered above. We have thus associated one charge value to each index $a$, which we can summarize in a vector $q^{[a]} := (2, 0, 0, -2)$. Using this definition, we can formulate the conservation of $S^z$ as a condition on the matrix elements:

$$H_{ab} = 0 \text{ if } q_a^{[a]} \neq q_b^{[a]}. \tag{45}$$

How does this generalize to tensors with a larger number of indices? To stay with the example, we can also write $H = \sum_{s_1 s_2 t_1 t_2} H_{s_1 s_2 t_1 t_2} |s_1\rangle |s_2\rangle \langle t_1| \langle t_2|$ as a tensor with 4 indices $s_1, s_2, t_1, t_2$ corresponding to the single-site basis $\{|s\rangle\} \equiv \{|\uparrow\rangle, |\downarrow\rangle\}$. The charge values $q^{[s]} = (1, -1)$ for this basis are obvious from the definition $q = 2S^z/\hbar$ (and the reason why we included the factor 2 in the rescaling). Since $S^z$ is additive, its conservation now implies that

$$H_{s_1 s_2 t_1 t_2} = 0 \text{ if } q_{s_1}^{[s]} + q_{s_2}^{[s]} \neq q_{t_1}^{[s]} + q_{t_2}^{[s]}. \tag{46}$$

Note that the indices corresponding to a ket appear on the left hand side of the inequality, while the ones corresponding to a bra appear on the right. For an arbitrary tensor, we therefore define one sign $\zeta = \pm 1$ for each leg, where we choose the convention $\zeta = +1$ ($\zeta = -1$) for a ket (bra); for the above example $\zeta^{[1]} = \zeta^{[2]} = +1$ for the first two indices $s_1, s_2$ and $\zeta^{[3]} = \zeta^{[4]} = -1$ for the legs of $t_1, t_2$. In diagrams, we can illustrate this sign by an arrow pointing into (for $\zeta = +1$) or out of (for $\zeta = -1$) the tensor, see Fig. 12.

Finally, we also introduce an offset $Q$, which we call the "total charge" of a tensor. The general **charge rule** for an arbitrary $n$-leg tensor $M$ then reads

$$\boxed{\forall a_1, a_2 \cdots a_n : \ \zeta^{[1]} q_{a_1}^{[1]} + \zeta^{[2]} q_{a_2}^{[2]} + \zeta^{[3]} q_{a_3}^{[3]} + \cdots + \zeta^{[n]} q_{a_n}^{[n]} \neq Q \ \Rightarrow \ M_{a_1 a_2 \cdots a_n} = 0} \tag{47}$$

Note that the signs $\zeta^{[i]}$ and the total charge $Q$ introduce some ambiguity: the charge rule (47) is still satisfied if we send $\zeta^{[j]} \to -\zeta^{[j]}$ and $q^{[j]} \to -q^{[j]}$ for some leg $j$, or if we send $\zeta^{[j]} q^{[j]} \to \zeta^{[j]} q^{[j]} + \delta Q$ and $Q \to Q + \delta Q$. However, introducing the signs and the total charge allows us to share the same $q$ vector between legs representing the same basis, e.g., all four legs of $H_{s_1 s_2 t_1 t_2}$ shared the same $q^{[s]}$. We can therefore fix the charge vectors $q$ of *physical* legs in the very beginning of the algorithm. Since also the signs $\zeta$ are fixed by conventions, for tensors with only physical legs one can solve the charge rule (47) for $Q$ (by inspecting which entries of a tensor are non-zero). Examples of this kind are given in Tab. 1.

On the other hand, if the total charge $Q$ and the charges $q^{[i]}$ of all but one leg $j$ of a tensor are fixed, one can also solve the charge rule (47) for the missing $q^{[j]}$:

$$\forall a_1, a_2 \cdots a_n : \ M_{a_1 a_2 \cdots a_n} \neq 0 \ \Rightarrow \ \zeta^{[j]} q_{a_j}^{[j]} = Q - \sum_{i \neq j} \zeta^{[i]} q_{a_i}^{[i]} \tag{48}$$

Table 1: Examples for charge definitions such that the tensors fullfill the charge rule (47). We consider spin-$\frac{1}{2}$ with $q = 2S^z/\hbar$, i.e., $q^{[s]} := (1, -1)$ for the single-site basis $\{\,|s\rangle\,\} \equiv \{\,|\uparrow\rangle, |\downarrow\rangle\,\}$ and $q^{[a]} := (2, 0, 0, -2)$ for the two-site basis $\{\,|a\rangle\,\} \equiv \{\,|\uparrow\uparrow\rangle, |\uparrow\downarrow\rangle, |\downarrow\uparrow\rangle, |\downarrow\downarrow\rangle\,\}$. The signs $\zeta$ are $+1$ ($-1$) for legs representing kets (bras). The total charge $Q$ can then be determined from the charge rule (47).

| Example | $\zeta^{[1]}$ | $q^{[1]}$ | $\zeta^{[2]}$ | $q^{[2]}$ | $\zeta^{[3]}$ | $q^{[3]}$ | $\zeta^{[4]}$ | $q^{[4]}$ | $Q$ |
|---|---|---|---|---|---|---|---|---|---|
| $H = \sum H_{s_1 s_2 t_1 t_2} \|s_1\rangle \|s_2\rangle \langle t_1\| \langle t_2\|$ | +1 | $q^{[s]}$ | +1 | $q^{[s]}$ | -1 | $q^{[s]}$ | -1 | $q^{[s]}$ | 0 |
| $H = \sum H_{ab} \|a\rangle \langle b\|$ | +1 | $q^{[a]}$ | -1 | $q^{[a]}$ | | | | | 0 |
| $S^z$ | +1 | $q^{[s]}$ | -1 | $q^{[s]}$ | | | | | 0 |
| $S^+$ | +1 | $q^{[s]}$ | -1 | $q^{[s]}$ | | | | | 2 |
| $S^-$ | +1 | $q^{[s]}$ | -1 | $q^{[s]}$ | | | | | -2 |
| $\|\uparrow\uparrow\rangle = \sum v_a \|a\rangle$ | +1 | $q^{[a]}$ | | | | | | | 2 |
| $\langle\uparrow\uparrow\| = \sum v_a^* \langle a\|$ | -1 | $q^{[a]}$ | | | | | | | -2 |
| $\|\uparrow\uparrow\rangle = \sum v_{s_1 s_2} \|s_1\rangle \|s_2\rangle$ | +1 | $q^{[s]}$ | +1 | $q^{[s]}$ | | | | | 2 |

This allows to determine the charges on the *virtual* legs of an MPS. As an example, let us write the singlet $|\psi\rangle = \frac{1}{\sqrt{2}} (|\uparrow\downarrow\rangle - |\downarrow\uparrow\rangle)$ as an MPS. The MPS in canonical form is given by

$$|\psi\rangle = \sum_{s_1 s_2, c} \Gamma^{[1]s_1}_{lc} \Lambda^{[1]}_c \Gamma^{[2]s_2}_{cr} |s_1\rangle |s_2\rangle \quad \text{with } \Lambda^{[1]} = \frac{1}{\sqrt{2}} \begin{pmatrix} 1 \\ 1 \end{pmatrix}, \tag{49}$$

$$\Gamma^{[1]\uparrow} = \begin{pmatrix} 1 & 0 \end{pmatrix}, \quad \Gamma^{[1]\downarrow} = \begin{pmatrix} 0 & 1 \end{pmatrix}, \quad \Gamma^{[2]\uparrow} = \begin{pmatrix} 0 \\ -1 \end{pmatrix}, \quad \Gamma^{[2]\downarrow} = \begin{pmatrix} 1 \\ 0 \end{pmatrix}. \tag{50}$$

Here, $l$ and $r$ are *trivial* indices $l \equiv r \equiv 1$, and only introduced to turn the $\Gamma^{[i]}$ into matrices instead of vectors. For trivial legs, we can (usually) choose trivial charges $q^{[\text{triv}]} := (0)$ which do not contribute to the charge rule. Moreover, we choose the convention that $\zeta = +1$ for left virtual legs, $\zeta = -1$ for right virtual legs and $Q = 0$, see Fig. 12(b). Then we can use the charge rule (48) of $\Gamma^{[1]}$ solved for $q^{[c]}$ and obtain:

$$\Gamma^{[1]\uparrow}_{11} \neq 0 \quad \overset{(47)}{\Rightarrow} \quad q^{[c]}_1 = 1, \qquad\qquad \Gamma^{[1]\downarrow}_{12} \neq 0 \quad \overset{(47)}{\Rightarrow} \quad q^{[c]}_2 = -1. \tag{51}$$

We use the *same* $q^{[c]} = (1, -1)$ for the left virtual leg of $\Gamma^{[2]}$; one can easily check that it also fulfills the charge rule (47) for $Q = 0$.

Strictly speaking, an operator with a non-zero total charge $Q$ does not preserve the charge of the state it acts on. However, it still preserves the block structure, because it changes the charge by exactly $Q$, e.g., $S^+$ increases it by 2. In contrast, $S^x$ (and similarly $S^y$) can both increases or decreases the charge, thus it can not be written as tensors satisfying eq. (47); only the combination $S^x_1 S^x_2 + S^y_1 S^y_2 = \frac{1}{2}(S^+_1 S^-_2 + S^-_1 S^+_2)$ preserves the charge. When writing $H$ as a charge conserving MPO, one can only use single-site operators with a well-defined $Q$.

## 5.2 Basic operations on tensors

Above, we motivated the form of the charge rule (47) and explained how to define the charges for various tensors. Thus, we can write both the initial state and the Hamiltonian in terms of tensors satisfying eq. (47). Now, we argue that tensor network algorithms require just a few basic operations on the tensors, namely (a) transposition, (b) conjugation, (c) combining two or more legs, (d) splitting previously combined legs (e) contraction of two legs, (f) matrix decompositions, and (g) operations on a single leg. These operations are depicted in Fig. 13. As we will show in the following, all of them can be implemented to preserve the charge rule

(a) $a \rightarrow \boxed{H} \rightarrow b$  $\longrightarrow$  $b \rightarrow \boxed{H} \rightarrow a$    (b) $a \rightarrow \boxed{H} \rightarrow b$  $\longrightarrow$  $a \leftarrow \boxed{\overline{H}} \leftarrow b$

(c) $\begin{smallmatrix} s_1 \\ s_2 \end{smallmatrix} \rightarrow \boxed{H} \rightarrow \begin{smallmatrix} t_1 \\ t_2 \end{smallmatrix}$  $\longrightarrow$  $a \rightarrow \boxed{H} \rightarrow b$    (d) $a \rightarrow \boxed{H} \rightarrow b$  $\longrightarrow$  $\begin{smallmatrix} s_1 \\ s_2 \end{smallmatrix} \rightarrow \boxed{H} \rightarrow \begin{smallmatrix} t_1 \\ t_2 \end{smallmatrix}$

(e) $a_1 \rightarrow \boxed{A} \rightarrow a_2$  $b_1 \rightarrow \boxed{B} \rightarrow b_2$  $\longrightarrow$  $a_1 \rightarrow \boxed{A} \rightarrow \boxed{B} \rightarrow b_2$

$d_1 \rightarrow \boxed{D} \rightarrow d_2$  $\longrightarrow$  $d_1 \rightarrow \boxed{D} \boxed{E}$
$\phantom{d_1 \rightarrow}\uparrow d_3$

(f) $l \rightarrow \boxed{M} \rightarrow r$  $\longrightarrow$  $l \rightarrow \boxed{U} \xrightarrow{c} \boxed{S} \xrightarrow{c} \boxed{V^\dagger} \rightarrow r$

(g) $f_1 \rightarrow \boxed{F} \rightarrow f_2$  $\longrightarrow$  $f_1 \rightarrow \boxed{F} \rightarrow \tilde{f}_2$ ,    $f_1 \rightarrow \boxed{F} \rightarrow f_2$  $\longrightarrow$  $f_1 \rightarrow \boxed{F} \rightarrow \boxed{S} \rightarrow f_2$

Figure 13: Basic operations required for tensor networks: (a) transposition, (b) conjugation, (c) combining two or more legs, (d) splitting previously combined legs (e) contraction of two legs, (f) matrix decompositions, and (g) operations on a single leg.

(47) and thus the block structure of the tensors. Thus, *any* algorithm using (only) these basic operations preserves the charges.

**Transposition** is by definition just a reordering of the legs. Clearly, (47) is then still valid if we reorder the charge vectors $q$ and signs $\zeta$ in the same way. Examples for the **conjugation** are already given in Tab. 1; beside the complex conjugation of the entries this includes exchanging bra and ket, i.e., a sign flip of all $\zeta$. The charge rule is then preserved if we also flip the sign of the total charge $Q$. For hermitian operators like $H$ the combination of complex conjugation and appropriate transposition changes neither the entries nor the charges of a tensor.

Another operation often needed is to **combine** two (or more) legs, e.g., before we can do an SVD, we need to view the tensor as a matrix with just two indices. In other words, we group some legs into a "pipe". The pipe looks like an ordinary leg, i.e., we define a sign $\zeta$ and charge vector $q$ for it. However, it has the internal structure that it consists of multiple smaller legs. Thus, we can later **split** it, e.g., after we did an SVD. For concreteness, let us again consider the above example $H_{s_1 s_2 t_1 t_2} \rightarrow H_{ab}$, i.e., we want to combine the indices $s_1, s_2$ into a pipe $a$ (and $t_1, t_2$ into a pipe $b$). In this case, we map the indices as $a(s_1, s_2) := 2s_1 + s_2$ and $b(t_1, t_2) := 2t_1 + t_2$. The charge rule is then preserved if we *define* the charge vectors $q$ of the pipes as $\zeta^{[a]} q^{[a]}_{a(s_1,s_2)} := \zeta^{[1]} q^{[s]}_{s_1} + \zeta^{[2]} q^{[s]}_{s_2}$ and $\zeta^{[b]} q^{[b]}_{b(t_1,t_2)} := \zeta^{[3]} q^{[s]}_{t_1} + \zeta^{[4]} q^{[s]}_{t_2}$, where $\zeta^{[1]} = \zeta^{[2]} = 1, \zeta^{[3]} = \zeta^{[4]} = +1$ are the signs of the indices $s_1, s_2, t_1, t_2$, and $\zeta^{[a]} = 1, \zeta^{[b]} = -1$ are the desired signs of the pipes. One can easily check that these definitions coincide with the previous ones, $q^{[a]} = (2, 0, 0, -2) = q^{[b]}$. Since the mapping of indices is one to one, one can also split a pipe into the smaller legs it consists of. However, note that this requires the $q$ vectors and signs $\zeta$ of these legs; the pipe should thus store a copy of them internally.

One of the most important (and expensive) operations on tensors is the **contraction** of legs. Let us consider two tensors $A_{a_1 a_2}$ and $B_{b_1 b_2}$ with charges $Q^A, q^{[a_i]}, \zeta^{A[i]}$ and $Q^B, q^{[b_i]}, \zeta^{B[i]}$, $i = 1, 2$. A contraction means to identify two indices and sum over it. Two indices can be identified if they represent the same basis, thus we *require* them to have the same charge vector $q$ and opposite signs $\zeta$. For example for the usual matrix product $C_{a_1 b_2} := \sum_c A_{a_1 c} B_{c b_2}$ we require $q^{[a_2]} = q^{[b_1]}$ and $\zeta^{A[2]} = -\zeta^{B[1]}$. The charge rule (47) for $C$ then follows from the charge rules of $A$ and $B$, if we define $Q^C := Q^A + Q^B$ and just copy the signs $\zeta$ and charge vectors $q$ for the free, remaining indices. Moreover, the cost of the contraction is reduced if

we exploit the block structure of $A$ and $B$, which becomes most evident if we have a block diagonal structure as in $H_{ab}$, eq. (44). On the other hand, we can also contract two legs of the same tensor, i.e., take a (partial) trace. The contributions of these two indices to the charge rule (47) then simply drop out and the rule again stays the same for the remaining indices of the tensor.

We collect linear algebra methods that take a matrix as input and decompose it into a product of two or three matrices under the name **matrix decomposition**. Examples include the diagonalization of a matrix $H = U^\dagger E U$, QR-decomposition $M = QR$ and SVD $M = USV^\dagger$. Here, we focus exemplary on the SVD, other decompositions can be implemented analogously. Let us first recap the properties of the SVD: it decomposes an arbitrary $m \times n$ matrix into a product $M_{lc} = \sum_c U_{lc} S_c (V^\dagger)_{cr}$, where $S_c$ are the $k = \min(m, n)$ positive singular values, and $U$ and $V$ are isometries, i.e., $U^\dagger U = \mathbb{1} = V^\dagger V$. The charge rule (47) for the matrix elements $M_{lc}$ implies a block structure: assuming that the basis states of the index $l$ are sorted by charge (which we discuss in the next paragraph), we can group indices with the same charge values together to form a block. Moreover, for each block of $l$ with a charge value $q_l^{[l]}$, there is at most one block of the index $r$ with compatible charges, i.e., we have some kind of pseudo block-diagonal form (even if the blocks are not strictly on the diagonal). Therefore, we can apply the SVD to each of the (non-zero) blocks separately and simply stack the results, which again yields a (pseudo) block-diagonal form for $U$ and $V^\dagger$ with the required properties. To define the charges of the new matrices we can ignore $S$, since it is only a trivial rescaling of one leg. Similar as for the contraction, we keep the charge vectors $q$ and signs $\zeta$ for the indices $l$ and $r$. Further, we choose the total charges as $Q^U := 0$ and $Q^V := Q^M$, as well as the sign $\zeta^{[c]}$ of the new index $c$ negative for $U$ and positive for $V$. The charge vector $q^{[c]}$ can then easily be read off using eq. (48), which yields $q^{[c]} := \zeta^{[l]} q^{[l]}$ (for both $U$ and $V^\dagger$).

Finally, the remaining operations needed for tensor networks are **operations on a single leg** of a tensor. One examples is a permutation of the indices of the leg, for example required to sort a leg by $q$ as mentioned above. Clearly, this preserves the charge rule if we apply the same permutation to the corresponding charge vector $q$. Simliarly, if we discard some of the indices of the leg, i.e., if we truncate the leg, we just apply the same truncation to the charge vector $q$. Lastly, we might also want to slice a tensor by plugging in a certain index of a leg, e.g., taking a column vector of a matrix. This requires to update the total charge $Q$ to preserve the charge rule, as one can show by viewing it as a contraction with a unit vector.

Above we explained how to define the charges for the $U(1)$ symmetry of charge conservation. In general, one can have multiple different symmetries, e.g., for spinfull fermions we might have a conservation of both the particle numbers and the magnetization. The generalization is straight-forward: just define one $q$ for each of the symmetries. Another simple generalization is due to another type of symmetry, namely $\mathbb{Z}_n$, where all the (in)-equalities of the charge rules are taken modulo $n$. An example for such a case is the parity conservation of a superconductor.

In TeNPy, the number and types of symmetries are specified in a `ChargeInfo` class [1]. We collect the $q$-vectors and sign $\zeta$ of a leg in a `LegCharge` class. The `Array` class represents a tensor satisfying the charge rule (47). Internally, it stores only the non-zero blocks of the tensor along with one `LegCharge` for each of its legs. If we combine multiple legs into a single larger "pipe" as explained above, the resulting leg will have a `LegPipe`, which is derived from the `LegCharge` and stores all the information necessary to later split the pipe.

All these classes can be found in the `tenpy.linalg.np_conserved`, which also contains functions for all the basic operations on tensors represented by an `Array` class, with an interface very similar to that of the NumPy (and SciPy) library [68]. Moreover, the module

`tenpy.networks.site` contains classes which pre-define the charges and local operators for the most commonly used models. For example the class `SpinHalfSite` defines the operators $S^+, S^-$, and $S^z$ (called Sp, Sm, and Sz) as instances of `Array`. The following code snippet uses them to generate and diagonalize the two-site hamiltonian (44); it prints the charge vector $q^{[a]}$ (by default sorted ascending) and the eigenvalues of $H$. A more extensive illustration of the interface can be found in the online documentation [1].

```
import tenpy.linalg.np_conserved as npc
from tenpy.networks.site import SpinHalfSite

site = SpinHalfSite(conserve="Sz")
Hxy = 0.5*(npc.outer(site.Sp, site.Sm) +
           npc.outer(site.Sm, site.Sp))
Hz = npc.outer(site.Sz, site.Sz)
H = Hxy + Hz
# here, H has 4 legs
H.iset_leg_labels(["s1", "t1", "s2", "t2"])
H = H.combine_legs([["s1", "s2"], ["t1", "t2"]], qconj=[+1, -1])
# here, H has 2 legs
print(H.legs[0].to_qflat().flatten())  # prints [-2  0  0  2]
E, U = npc.eigh(H)
print(E)   # prints [ 0.25 -0.75  0.25  0.25]
```

# 6 Conclusion

In these lecture notes we combined a pedagogical review of MPS and TPS based algorithms for both finite and infinite systems with the introduction of a versatile tensor library for Python (TeNPy) [1]. While there exists by now a huge arsenal of tensor-product state based algorithms, we focused here on the time evolving block decimation (TEBD) [24] and the density-matrix renormalization group (DMRG) method [8]. For both algorithms, we provided a basic introduction and showed how to call them using the TeNPy package. Let us stress that there are further tricks and tweaks to improve the accuracy of the results. Beside tuning the different algorithm parameters, for which we refer to documentation of the package [1], one can for example extrapolate the results in the bond dimension to $\chi \to \infty$ [11, 69].

The TeNPy package contains some minimal working codes for finite as well as infinite TEBD and DMRG algorithms based on standard Python libraries. These "toy codes" are intended as a pedagogical introduction, to give a flavor of how the algorithms work.

While we did not cover genuine 2D tensor-product state methods, we note that the tensor tools build into TeNPy allow for a simple implementation of general tensor networks in higher dimensions as well. In particular, the method of conserving abelian symmetries discussed in the previous chapter directly carries over to 2D tensor-product states. Several complementary approaches for the simulation of higher dimensional tensor networks are currently under development [40].

We close these notes with a comment on the efficiency of the latest TeNPy library (version 0.3.0) [1] by comparing its speed with the ITensor C++ library (version 2.1.1), https://itensor.org. For simulations involving MPS with a small bond dimension $\chi_{\max}$, we find that the TeNPy library suffers from a large overhead of the Python code. Yet, for simulations of MPS with a larger bond dimensions $\chi_{\max}$, both libraries spend nearly all of the CPU time in linear algebra routines using the underlying BLAS/LAPACK libraries, such that the Python overhead becomes negligible. For example, we find that the application of the TEBD algorithm to time evolve an MPS with $\chi_{\max} = 100$ takes the same time with TeNPy as with ITensor, when no quantum numbers can be exploited. When the $S^z$ conservation of a spin-$\frac{1}{2}$ chain is used, TeNPy reaches nearly the same speed as ITensor for MPS with bond dimensions $\chi_{\max} \gtrsim 300 - 350$.

For the DMRG algorithm, a direct comparison is difficult since the libraries use different eigensolvers; we find however that the ITensor library is generally faster than the current TeNPy. Contributions to the TeNPy library are very welcome!

# Acknowledgements

We are grateful to Roger Mong and Michael Zaletel for stimulating discussions. We acknowledge contributions to the TeNPy package by others, including Maximilian Schulz, Leon Schoonderwoerd, and Kévin Héméry; the full list of contributors is distributed with the code.

**Funding information**   FP acknowledges the support of the DFG Research Unit FOR 1807 through grants no. PO 1370/2-1, TRR80, the Nanosystems Initiative Munich (NIM) by the German Excellence Initiative, and the European Research Council (ERC) under the European Union's Horizon 2020 research and innovation program (grant agreement no. 771537).

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
