# Peer review of "Efficient numerical simulations with Tensor Networks: Tensor Network Python (TeNPy)"

_SciPost Physics Lecture Notes, doi:SciPost Phys. Lect. Notes 5 (2018)_

## Round 1 · Referee Report · Claudius Hubig (Referee 1) · 2018-5-21

Strengths

  1. excellent introduction into matrix-product states and their underlying physical motivation

  2. good description of the TeNPy library with example codes provided both here and in the documentation accompanying TeNPy.

  3. complete description for the implementation of Abelian symmetries

  4. discussion of basic TEBD and DMRG methods as provided by TeNPy

Weaknesses

  1. reliance on the Γ-Λ-notation which is not necessary in the presented applications and numerically unstable

  2. the discussion of DMRG is relatively short and not describing the current state of the art

Report

The lecture notes both provide an excellent motivation for and description of matrix-product states as well as an introduction into the presented TeNPy Python package. The Python package itself is excellently documented (outside these lecture notes), provides the examples described in the lecture notes and appears to provide reasonable performance given the design goals which focus on usability, ease of extensibility and understandability.

Modulo some minor typos and small extensions (see below), my main concerns with the lecture notes submitted are then two-fold. As soon as these are addressed in some way, I wholeheartedly suggest publication of these very readable, pedagogical and thorough lecture notes.

First, the algorithms are presented using the Γ-Λ-notation/gauge for matrix product states. While reasonably straightforward, restoring this gauge after tensor updates involves the numerically unstable multiplication by the inverse singular value tensor. However, neither for the presented TEBD algorithm nor for DMRG updates is it necessary to keep the state within the Γ-Λ notation and indeed, in the section on DMRG, a different gauge is used: the much more stable alternative, the mixed canonical form with left-orhogonalised tensors (A), an orthogonality centre (either a single-site tensor M or a bond tensor C) and right-orthogonalised tensors (B).
The Γ-Λ notation is primarily useful when a) updating site tensors in parallel in TEBD (cf. Urbanek and Soldan, 2016) or b) evaluating many local expectation values. While a) is a potentially useful application, Urbanek and Soldan used an interesting trick to avoid the multiplication by the inverse singular value tensor. b) is potentially applicable, but evaluating expectation values is typically much cheaper than optimising or time-evolving tensors, so it appears unreasonable to enforce the Γ-Λ gauge also during the intermediate operations (instead of restoring it at the end). If the Γ-Λ notation provides other benefits, it would be useful to present those benefits in section 3.1 or instead only use the A-M-B form throughout the entire lecture notes.

Second, the section on DMRG should be extended to capture more of the current state of the art. In particular, the single-site update is missing completely. The primary aim of the density matrix perturbation or the subspace expansion is not so much to improve the convergence of the 2-site DMRG (2DMRG), but also to allow for a reliable single-site DMRG (1DMRG) update with its associated computational speed-up. The relative advantages of 2DMRG (slightly better convergence of other observables, less potential to get stuck) and 1DMRG (much faster, larger bond dimensions, often better energies) should be discussed in a pedagogical introduction to variational ground-state searches with MPS. A ‘staging scheme’ to incrementally increase the bond dimension and accuracy of the calculation during the sweeps is also present in the example code (I think, the example is not entirely clear) but not explained in detail. Further methods to avoid convergence problems (multi-grid DMRG, perturbation Hamiltonians, initialisation of the starting state etc.) could also be mentioned. Finally, extrapolations to the limit $\chi \to \infty$ are an essential part of large-scale DMRG calculations and should not be excluded from a pedagogical introduction.

Requested changes

  1. Reconsidering the use of Γ-Λ gauge at least in the pedagogical introduction. If keeping it, explaining the trick used by Urbanek and Soldan (Computer Physics Communications, 2016) to avoid multiplication by the inverse singular value tensor would be useful.

  2. Extending the section on DMRG to present more of the details of the algorithm and how to implement/use those within TeNPy would be worthwhile.

Additionally, I’d like to suggest the following minor corrections:

  1. "the Hadane phase in quantum spin…" should read "the Haldane phase…".

  2. "For each algorithm, we give a short example code show how to call" should read either "showing how to" or "to show how to call".

  3. Below equation (1), ${ | \alpha \rangle_{L(R)} }$ are called an "orthonormal basis" of $\mathcal{H}_L$. A basis is a complete set of vectors spanning the entire vector space, which cannot be since $\mathcal{H} = \mathcal{H}_L \otimes \mathcal{H}_R$ does not ensure that the left and right spaces have the same dimension. Instead, the ${ | \alpha \rangle_{L(R)} }$ are the left and right Schmidt bases and form an orthonormal basis of a subspace of $\mathcal{H}_L$ and $\mathcal{H}_R$ respectively.

  4. In section 3.1, it may be useful to also introduce the adjective "auxiliary" to describe the bond/virtual indices, as it is commonly encountered in the literature.

  5. I’m not sure it is a good idea to describe an MPS on a finite system as "uniform" (above equation (9)). While strictly speaking true, it may be more appropriate to reserve the adjective "uniform" to infinite-system MPS.

  6. In section 3.3, imaginary time evolution is described as "being used to find ground states of the Hamiltonian" (eq. (22) and the sentence above it). However, imaginary TEBD is not the method of choice for this task (as also explained later on) and it may confuse a novice reader. A more appropriate example application of imaginary time evolution may be the evaluation of Green’s functions.

  7. At the bottom of page 14, it is said that the TEBD method can only be applied to nearest-neighbour Hamiltonians. The introduction of swap gates (Stoudenmire and White, 2010), however, avoids this problem entirely at only slightly larger computational effort.

  8. The example code for TEBD with TeNPy is split over a line-break, which makes copy-pasting it a bit hard. In general, a more traditional formatting with a monospaced typeface may be worth considering (also depending on the editor’s preferences).

  9. On page 16, Ref. 61 is mentioned to describe finite state machines for MPO construction. A very readable other paper on that topic is Paeckel et al 2017 (SciPost Phys. 3, 035), which is certainly worth mentioning here.

  10. I’m not sure what is meant by "even if they might not be the true Schmidt basis" in footnote 3 on page 17. The DMRG procedure always keeps a Schmidt decomposition between the current left- and right environment and hence always keeps a left and right Schmidt basis.

  11. Fig. 8b) is slightly misleading (as then also mentioned in the text). While the effective Hamiltonian does take this form, it is very suboptimal to compute it. Instead, the authors might consider showing $\psi_{\mathrm{eff}}$ (i.e. the two-site tensor) and the product $\hat H_{\mathrm{eff}} \cdot \psi_{\mathrm{eff}}$, which would avoid the impression that one has to explicitly evaluate $\hat H_{\mathrm{eff}}$. As an aside, in the case of quantum numbers which split the local state-space into one-dimensional blocks (i.e. a single state per quantum number), the ordering L·W·M·R instead of L·M·W·R allows precomputation of L·W prior to all Lanczos iterations and is typically somewhat faster than the asymptotically-optimal ordering, at least in 1DMRG calculations.

  12. On page 19, it is mentioned that the left and right bases have to be orthonormal to ensure optimal truncation. However, these bases have to be orthonormal for the much more fundamental reason that one wishes to update the tensor using a standard instead of a generalised eigenvalue problem (as eg. explained in Schollwöck 2011, around Eq. 208). This fact should not be omitted!

  13. On page 21, it may be worthwhile to mention the recent VUMPS algorithm/method as an alternative to the previous iMPS methods.

  14. Section header "4.1 Infinte Time Evolving…" should read "4.1 Infinite Time Evolving…".

  15. In footnote 5, the authors say that they restrict themselves to "local, abelian symmetries". In fact, the symmetries considered are global symmetries (i.e., not local/gauge symmetries). The restriction on locality is that the symmetry has to act locally; e.g. momentum conservation on a ring in real space cannot be implemented in the setting described here, one has to go to momentum space to do so.

  16. I am not sure I understand the role of the total charge Q per tensor completely, in particular during tensor-tensor contractions. Do I understand correctly that Q acts as an additional one-dimensional dummy index carrying a total charge out of/into the tensor, with those legs combined into a new "pipe" leg during tensor-tensor contractions? If so, it may be worthwhile to offer this alternative explanation as well.

  17. In the installation instructions (doc/INSTALL.rst) of the TeNPy package on Github, the probably aliased command "git co <tag>" is used. The authors may wish to replace that by "git checkout <tag>".

  18. Finally, I would suggest to link to the excellent documentation of the TeNPy package at https://tenpy.github.io/ directly from the paper. It is absolutely worth reading and hence also pointing out to readers.

  • validity: high
  • significance: high
  • originality: high
  • clarity: top
  • formatting: good
  • grammar: excellent

Author:  Johannes Hauschild  on 2018-08-30  [id 312]

(in reply to Report 1 by Claudius Hubig on 2018-05-21)
Category:
answer to question

We thank you very much for the positive and detailed report and are grateful for the effort that went into it.

On the first point: We believe that the $\Gamma$-$\Lambda$ notation is very useful from a pedagogical point of view to understand the canonical form. In particular it provides a simple understanding of the relations between the left, right, and mixed canonical form - as it allows to read off the Schmidt decomposition of $|\psi\rangle$ at each bond at the same time. We are aware that using the $\Gamma$ matrices in the code leads to numerical instabilities. However, we consider these instabilities as a minor detail of the implementation, less important than understanding the overall concepts of TEBD and DMRG. To the best of our knowlege, the trick for the TEBD used by Urbanek and Soldan was originially introduced by Hastings in 2009 (our Ref. 59), which we referenced in the paragraph before the TEBD example code (Of course, the trick is used in the TeNPy implementation)

On the second point: The goal of the lecture notes is to provide an accessible introduction for the newcommers to the field. We feel that explaining too many details about the different parameters in these notes will lead to unnecessary confusions, and instead refer to the documentation for that, which will be kept up to date with the implementation. We extended the discussion of 2DMRG vs 1DMRG sligthly (now before the example code to have it closer to the algorithm) and added a sentence about the extrapolation $\chi \rightarrow \infty$ in the conclusion.

On the requested changes:

1.,2.: See above

  1. Corrected.

  2. Corrected.

  3. We see the point but do not fully agree with this comment: we can always complete the used left and right Schmidt-Bases to a full basis of $\mathcal{H}_{L(R)}$. Therefore, we changed the text to "orthonormal basis of (the relevant subspaces of) $\mathcal{H}_L$ ($\mathcal{H}_R$)".

  4. Added.

  5. Agree, corrected.

  6. Added the Green's functions as example and a footnote that DMRG is better.

  7. Added a footnote mentioning the swap gates.

  8. Changed/Improved.

  9. Added.

  10. Say we optimize on sites (n, n+1) and find a new optimal state, then e.g. the $|a_{n+2}\rangle_R$ (in the sense of Fig. 4c) are still orthonormal, but not the correct Schmidt basis with respect to the new state. Yet, we agree that the formulation was misleading and changed it.

  11. Like for the $\Gamma$-$\Lambda$ notation, we think that it is helpful from a pedagogical point of view to show the network of $H_{eff}$, even if it gets never fully contracted in an efficient DMRG implementation.

  12. Included this fact.

  13. We added an paragraph to mention VUMPS at the end of the chapter on iDMRG.

  14. Changed.

  15. Corrected.

  16. Yes, one could basically think of it like a dummy index with an extra charge. For contractions, we do not necessarily need to combine legs into pipes, but yes, for and SVD you can think of it like that. The total charge is introduced to be able to write e.g. S+ and S- as matrices with just two indices, and all legs sharing the same charge vector q (defined only once for a "physical" leg).

  17. Thank you for this hint, corrected.

  18. Agree, added.

---

## Round 1 · Referee Report · Anonymous (Referee 2) · 2018-7-4

Strengths

1) Well-written introduction, very pedagogical

2) Examples of codes for python TN libraries

3) Discussion about abelian symmetries in the TN libraries

Weaknesses

1) Only focused on Matrix Product States

2) No numerical benchmarks of the libraries

3) No discussion of non-abelian symmetries

Report

In this paper the authors offer a pedagogical introduction to some basic aspects of tensor network states and many-body entanglement theory, focusing their explanations on Matrix Product States (MPS) and their associated methods. The authors also explain the implementation of abelian symmetries in MPS, and provide a practical guide for their tensor network library in Python (TeNPy) with practical codes.

This is definitely a very useful paper, which is also nicely written It definitely deserves publication in SciPost. I have some observations though, that the authors should consider before publication:

1) Could the authors comment also (at least briefly) on the implementation of non-abelian symmetries?

2) The authors say in the title that they discuss “tensor networks”. However, in practice they only discuss MPS. I would therefore either discuss briefly other tensor networks in their paper (PEPS, MERA…), or modify the title and say “Matrix Product States” instead of Tensor Networks.

3) The paper is fully analytical, and no numerical calculation is shown. Since this is a paper about some programming libraries, it would be very helpful to the reader if the authors showed some examples of numerical benchmarks with TeNPy.

Requested changes

1) Comment on non-abelian symmetries

2) Modify title or discuss TN beyond 1d

3) Add numerical benchmarks

  • validity: high
  • significance: good
  • originality: good
  • clarity: high
  • formatting: perfect
  • grammar: perfect

Author:  Johannes Hauschild  on 2018-08-30  [id 313]

(in reply to Report 2 on 2018-07-04)

We thank the referee for the positive report and would like to address his points below:

  1. Actually, we did shortly comment on non-abelian symmetries in the footnote on page 25. To make it more prominent, we moved it into the main text right before section 5.1 in the revised script.

  2. These lecture notes introduce the TeNPy library, which is what the title is focused on. We called the library TeNPy, because it will most likely contain codes on 2D tensor networks in future versions. While we only discuss the 1D MPS methods as examples of general tensor networks, the chapter 5 on abelian symmetries applies to tensor networks in 2D as well. The abstract clearly says what we discuss, so we feel that the title is justified.

  3. We have include a paragraph on the performance of the TeNPy library at the end of the conclusions. High performance is only a secondary goal of the library, and in comparison to other C++ libraries like ITensor only reached for simluations involing large bond dimensions. Yet, we hope and believe that the TenPy library proves useful to the community due to its accessibility and flexibility - in particular also for the audience targeted by these lecture notes. Moreover, we hope that we can still optimize a few parts of the library in future versions to increase its performance, which would render a detailed benchmark in these lecture notes out of date.

---

## Round 2 · Referee Report · Claudius Hubig · 2018-9-12

Strengths

5. Compared to the first version, the description of DMRG and TEBD is more complete and now adequately summarises the current state of the art together with pointers to further reading.

Points 1. through 4. carry over from the first version of the paper.

Report

While I still believe that the Γ-Λ notation is suboptimal in practical applications, I understand the authors position regarding its higher pedagogical value and agree with this assessment. The authors have additionally introduced pointers towards further reading regarding TEBD and DMRG implementations and methods and have implemented other minor problems/typos found in the first version.

In the updated lecture notes, I found two more minor issues; these, however, do not detract from the overall quality of the notes. I hence suggest publication of the updated lecture notes.

Requested changes

The authors may wish to consider the following two points:

1. On page 4, Section 1, last paragraph: "review of the basics MPS and TPS based" should read "review of the basic MPS and TPS based"

2. I am confused by footnote 2 on pg. 11: First, it may make more sense to move it to the TEBD chapter as it deals primarily with TEBD; second, regardless of whether Schmidt values at other bonds are preserved or not during a truncation, a local truncation can change the energy of the state and third, the canonical form can easily be reinstated during a TEBD calculation and hence loss of this form can not be the sole reason TEBD does not preserve the energy. May I suggest deleting the footnote and moving its last sentence about TDVP to the TEBD chapter? E.g. in the paragraph after Eq. 32, one may add a sentence that TEBD does not necessarily preserve the energy due to its evolve-then-truncate approach and that an alternative exists in the form of TDVP.

---

## Round 2 · Referee Report · Anonymous · 2018-9-13

Report

The authors have properly addressed the concerns from my previous review. Therefore I think that the paper is ready for publication.

---

## Round 2 · Author Response

Dear Editor,

we thank you for communicating the reports and are grateful for having recieved such detailed and positive reviews. We have carfully considered the comments and requested changes of the referees and would like to address their main points in the following.

First Report:

1) We believe that the $\Gamma$-$\Lambda$ notation is very useful from a pedagogical point of view to understand the canonical form. In particular it provides a simple understanding of the relations between the left, right, and mixed canonical form--as it allows to read off the Schmidt decomposition of $|\psi\rangle$ at each bond at the same time. We are aware that using the $\Gamma$ matrices in the code leads to numerical instabilities. However, we consider these instabilities as a minor detail of the implementation, less important than understanding the overall concepts of TEBD and DMRG. To the best of our knowlege, the trick for the TEBD used by Urbanek and Soldan was originially introduced by Hastings in 2009 (our Ref. 59), which we referenced in the paragraph before the TEBD example code (Of course, the trick is used in the TeNPy implementation)

2) The goal of the lecture notes is to provide an accessible introduction for the newcommers to the field. We feel that explaining too many details about the different parameters in these notes will lead to unnecessary confusions, and instead refer to the documentation for that, which will be kept up to date with the implementation. We extended the discussion of 2DMRG vs 1DMRG sligthly (now before the example code to have it closer to the algorithm) and added a sentence about the extrapolation $\chi \rightarrow \infty$ in the conclusion.

Second Report:

  1. Actually, we did shortly comment on non-abelian symmetries in the footnote on page 25. To make it more prominent, we moved it into the main text right before section 5.1 in the revised script.

  2. These lecture notes introduces the TeNPy library, which is what the title is focused on. We called the library TeNPy, because it will most contain codes on 2D tensor networks in future versions. While we only discuss the 1D MPS methods as examples of general tensor networks, the chapter 5 on abelian symmetries applies to tensor networks in 2D as well. The abstract clearly says what we discuss, so we feel that the title is justified.

  3. We have included a paragraph on the performance of the TeNPy library at the end of the conclusions. High performance is only a secondary goal of the library, and in comparison to other C++ libraries like ITensor only reached for simluations involing large bond dimensions. Yet, we hope and believe that the TenPy library proves useful to the community due to its accessibility and flexibility -- in particular also for the audience targeted by these lecture notes. Moreover, we hope that we can still optimize a few parts of the library in future versions to increase its performance, which would render a detailed benchmark in these lecture notes out of date.

---

## Round 2 · List of Changes

* We followed most of the requested changes of the referees as detailed above and in the first replies to the reports 1 and 2.
* Corrected the figure reference between eqs. (28-29).
* Corrected indices in eqs. (31-32).
* Changed the sign for the field-term in the Hamlitonian (34) to follow the usual convention.
Adjusted the following eqs. (36-37) and the example code accordingly.
* Added a comment that 'hz' can be given as an numpy array for site dependent fields in the model example code on page 16.
* Adjusted the typesetting of the code examples.
* Adjusted example codes to use double quotes for strings throughout.
* Corrected Ref. [9].
* Added Refs. [33, 61, 64, 67, 69].
* Corrected further minor typos.

---

## Editorial Decision

published